# LEARNING DISENTANGLEMENT IN AUTOENCODERS THROUGH EULER ENCODING

## ABSTRACT

Noting the importance of factorizing (or disentangling) the latent space, we propose a novel, non-probabilistic disentangling framework for autoencoders, based on the principles of symmetry transformations that are independent of one another. To the best of our knowledge, this is the first deterministic model that is aiming to achieve disentanglement based on autoencoders without pairs of images or labels, by explicitly introducing inductive biases into a model architecture through Euler encoding. The proposed model is then compared with a number of state-of-the-art models, relevant to disentanglement, including symmetry-based and generative models based on autoencoders. Our evaluation using six different disentanglement metrics, including the unsupervised disentanglement metric we propose here in this paper, shows that the proposed model can offer better disentanglement, especially when variances of the features are different, where other methods may struggle. We believe that this model opens several opportunities for linear disentangled representation learning based on deterministic autoencoders.

## 1 INTRODUCTION

Learning generalizable representations of data is one of the fundamental aspects of modern machine learning (Rudin et al., 2022). In fact, better representations are more than a luxury now, and is a key to achieving generalization, interpretability, and robustness of machine learning models (Bengio et al., 2013; Brakel & Bengio, 2017; Spurek et al., 2020). One of the primary and desired characteristics of the learned representation is factorizability or disentanglement, so that latent representation can be composed of multiple, independent generative factors of variations. The disentanglement process renders the latent space features to become independent of one another, providing a basis for a set of novel applications, including scene rendering, interpretability, and unsupervised deep learning (Eslami et al., 2018; Iten et al., 2020; Higgins et al., 2021).

Deep generative models, particularly that build on variational autoencoders (VAEs) (Kingma & Welling, 2013; Kumar et al., 2017; Higgins et al., 2017; Tolstikhin et al., 2018; Burgess et al., 2018; Chen et al., 2018; Burgess et al., 2018; Kim & Mnih, 2018; Zhao et al., 2019), have shown to be effective in learning factored representations. Although these approaches have advanced the disentangled representation learning by regularizing the latent spaces, there are a number of issues that limit their full potential: (a) VAE-based models consist of two loss components, and balancing these loss components is a well known issue (Asperti & Trentin, 2020) (b) it is almost impossible to honor the idealized notion of having a known prior distribution for VAEs in practical settings (Takahashi et al., 2019; Asperti & Trentin, 2020; Zhang et al., 2020; Aneja et al., 2021) and, (c) factorizing the aggregated posterior in the latent space does not guarantee corresponding uncorrelated representations (Locatello et al., 2019). An alternative approach for achieving disentangled representations is through seeking irreducible representations of the symmetry groups (Cohen & Welling, 2014; Higgins et al., 2018; Painter et al., 2020; Tonnaer et al., 2022), where the aim is to find latent space transformations that are independent of one another, underpinned by well-defined mathematical framework(s) based on group theory. As this group of methods exploits the notion of transitions between samples, they require pairs of images representing the transitions (Cohen & Welling, 2014; Painter et al., 2020) or equivalent labels (Tonnaer et al., 2022). Regardless of the approach, as shown in Locatello et al. (2019), it is fundamentally impossible to learn disentangled representations without having inductive biases on either the model or the dataset, and both VAE- and symmetry-based approaches exemplify implicitly embedding inductive bias.

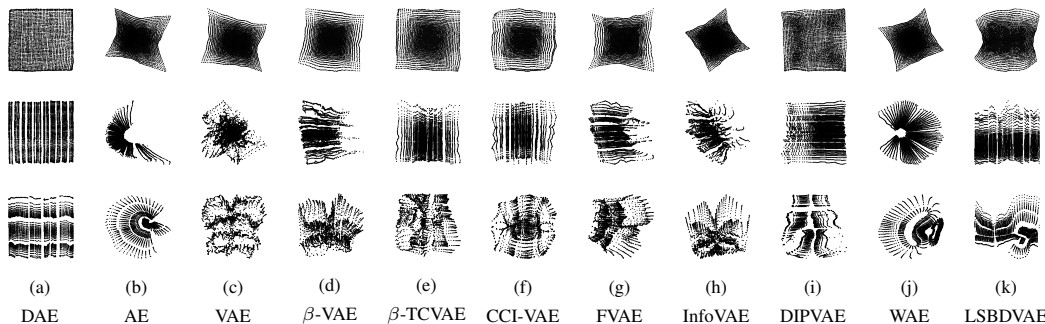

(a) DAE    (b) AE    (c) VAE    (d) $\beta$-VAE    (e) $\beta$-TCVAE    (f) CCI-VAE    (g) FVAE    (h) InfoVAE    (i) DIPVAE    (j) WAE    (k) LSBDVAE

Figure 1: Latent spaces learned by different models. Ideal learned latent space should cover a two-dimensional grid (Higgins et al., 2018). The first, second and third rows show the latent spaces learned from three datasets, namely, $XY$, 2D Arrow, and 3D Airplane datasets, respectively. Columns correspond to different models stated at the bottom of every column. It can be seen that the proposed model, DAE, achieves the best disentanglement.

Despite the advances, we note a number of issues in the existing approaches about how they address disentanglement. Firstly, the majority of the VAE-based approaches are probabilistic, and as such, the quality of the disentanglement depends on ideal or near-ideal priors, and on the process of learning the correct posteriors for a given data. Secondly, the majority of symmetry-based disentangling approaches need pairs of images or labels, even in the unsupervised setting, owing to the requirements around inductive bias. Thirdly, none of these models conform to the formal definition of a linear disentangled representation proposed by Higgins et al. (2018). Finally, and most importantly, none of the existing approaches have the unsupervised approach for introducing inductive biases (required for disentanglement) both on the models and on the datasets, essentially demanding labels or image pairs. Motivated by these shortcomings, in this paper, we propose a novel approach for deriving disentangled representation learning, with the following key contributions: We,

- propose a totally unsupervised approach for introducing inductive bias into the model and data, without requiring pairs of images or labels,

- propose a non-probabilistic approach that does not involve any priors or learning posteriors,

- ensure that our approach conforms to the formal definition of a linearly disentangled representation as defined in Higgins et al. (2018),

- propose a new unsupervised metric, namely, Grid Fitting Score (GF-Score), to quantify the disentanglement, echoing the aspiration of an ideal disentanglement measure outlined in Higgins et al. (2018), and

- demonstrate the implementation of a formally defined disentanglement using autoencoders.

As such, the proposed approach, which we name Disentangling Auto-Encoder (DAE), offers a theoretically sound framework for learning independent multi-dimensional vector subspaces, and hence towards learning disentangled representations. To the best of our knowledge, this is the first attempt to actually implement a disentanglement approach using deterministic autoencoders, especially without pairs of images or labels, and hence in a truly unsupervised manner. We provide a glimpse into the capability of the proposed model for disentanglement using three datasets compared against ten other models, which are either autoencoder-based probabilistic models or symmetry-based disentangled models, that do not require any labels or pairs of inputs in Figure 1.

The rest of this paper is organized as follows. In Section 2 we review the related work, focusing on VAE-based and symmetry-based approaches. This is then followed by a derivation of AE-based non-probabilistic approach for deriving disentangled representations in Section 3. In Section 4, we perform a detailed evaluation to decide the overall performance of the proposed model, using ten baseline models, six datasets, and six disentanglement metrics, and discuss our findings. We then conclude the paper in Section 5 with directions for further research. Given the space constraints, we highlight the prominent results in the main part of the paper, while providing the remaining set of results and relevant material as part of the Appendix.

## 2 RELATED WORK

### 2.1 DISENTANGLEMENT

Disentangled representation learning focuses on learning a set of independent factors containing useful but minimal information for a given task, such that their variations are orthogonal to each other while accounting for the entire dataset (Bengio et al., 2013; Higgins et al., 2018). This essentially entails a method or a set of methods for decoupling correlations between latent variables. A large body of work around disentanglement, and the ideal properties of a disentangled representation can be found in Ridgeway (2016); Eastwood & Williams (2018); Ridgeway & Mozer (2018); Zaidi et al. (2020). Among a number of desirable properties of a disentangled representation, modularity, compactness and explicitness are three critically important properties. A number of metrics have been proposed in the literature to quantify these properties (Higgins et al., 2017; Kim & Mnih, 2018; Eastwood & Williams, 2018; Chen et al., 2018; Do & Tran, 2019; Sepliarskaia et al., 2019). In our work, we use the notions outlined in Zaidi et al. (2020), where the metrics are divided into three classes, namely, Intervention-based, Predictor-based, and Information-based metrics. These metrics are all used in a supervised manner and can be of indicators for the robustness of the representation to noise, and for the non-linearity of the relationships between learnt representations and ground truth factors, in addition to the three properties outlined above.

### 2.2 AUTOENCODER-BASED PROBABILISTIC MODELS

AE-based probabilistic generative models are realized by replacing the conventional encoder $E_\phi$ and decoder $D_\theta$ with probabilistic counterparts. The probabilistic encoder, denoted by $q_\phi(z|x)$, is used to approximate the intractable true posterior, and the probabilistic decoder, denoted by $p_\theta(x|z)$, is used to reconstruct the $x$ from $z$ (Kingma & Welling, 2013). The majority of the previous work on disentangled representation learning are based on probabilistic models, particularly building on VAE. They enforce regularization in the latent space that either regularizes the approximate posterior $q_\phi(z|x)$ or the aggregate posterior $q(z) = \frac{1}{N} \sum_{i=1}^{N} q_\phi(z|x^{(i)})$, as summarized in Tschannen et al. (2018). The overall objective of the majority of the VAE-based methods can be expressed as:

$$L_{recon}(\phi, \theta) + L_{reg}(\phi) \tag{1}$$

where $L_{reg}(\phi)$ is a regularizer of the concerned generative model. A carefully designed regularizer should enable the model to achieve better disentanglement, either by controlling the capacity of the latent space, or by measuring the total correlation between latent variables. However, it is worth noting that factorizing aggregated posterior using regularizers does not guarantee linear disentangled representations (Locatello et al., 2019). We summarize the regularization terms of seven state-of-the-art generative models in Appendix A (See Columns 2 and 3 of Table 3).

### 2.3 SYMMETRY-BASED DISENTANGLING MODEL

While Higgins et al. (2018) proposed a formal definition of linear disentangled representations, it was generic, so that no specific architecture, model or technique were defined. As such, it does not provide an actual mechanism for learning such disentangled representations, albeit providing a formal definition, which is essential for this purpose. From the definitions in Higgins et al. (2018), a symmetry group can be decomposed as a product of multiple subgroups, if suitable subgroups can be identified. This can provide an intuitive method for disentangling the latent space, if subgroups that independently act on subspaces of a latent space can be found. If actions applied on each of the subgroups affect only the corresponding subspace, these actions are called disentangled group actions. In other words, disentangled group actions only change a specific property of the state of an object, and leaves the other properties invariant. If there is a transformation in a vector space of representations, corresponding to a disentangled group action, the representation is called a disentangled representation.

The concept and implementation of symmetry-based disentangled representations were proposed using pairs of images in Cohen & Welling (2014). However, owing to the limitation around commutative Lie groups, upon which this model is built upon, the real world applicability of the technique from Cohen & Welling (2014), especially across a range of diverse datasets, are limited. Following

a formal definition for linear disentangled representations in Higgins et al. (2018), there has been a considerable amount of effort to learn the transitions between images (Caselles-Dupré et al., 2019; Quessard et al., 2020; Painter et al., 2020). The transitions between images are learned by treating each transition as a sequence of transitions until the base transition relies on pairs of images and by using additional networks. An alternative approach is to rely on labels, for example, as in Tonnaer et al. (2022), where they propose two Diffusion VAE-based methods (Rey et al., 2019), namely semi-supervised and unsupervised, along with a new metric called LSBD (Linear Symmetry-Based Disentanglement metric). The former model relies on labels, while the latter does not. As such, the latter model is directly relevant to our work, and, we use this as one of the baselines for our evaluation (See Section 4).

## 3  FRAMEWORK FOR DAE

The deterministic, and hence, non-probabilistic, approach we propose here, builds on the autoencoder architecture (rather than variational autoencoders). We provide the relevant background on the disentangled representations from Higgins et al. (2018) in the Appendix A.2. In this section, we define necessary mathematical framework and a corresponding neural network architecture implementing the proposed disentangling autoencoder.

### 3.1  ASSOCIATION BETWEEN THE DISENTANGLED REPRESENTATION AND AUTOENCODER

The definition of disentangled group actions from Higgins et al. (2018) assumes that a group $G$ can be decomposes a direct product $G = G_1 \times \cdots \times G_n$. To relax the condition, we consider a group $G$, which is generated by $S = \{s_1, s_2, ..., s_n\}$ subject to a set of $R$ of relations among elements in $S$. Let $W$ be a set of world-states and suppose we have a group action $\cdot : G \times W \to W$. Then, we say that the action is disentangled by the relation $R$ if there is a decomposition $W = W_1 \times \cdots \times W_n$ and actions $\cdot_i :< s_i > \times W_i \to W_i$, $i \in \{1, ..., n\}$ such that:

1.  $(s_1^{\epsilon_1}, ..., s_n^{\epsilon_n}) \cdot (w_1, ..., w_n) = (s_1^{\epsilon_1} \cdot w_1, ..., s_n^{\epsilon_n} \cdot w_n)$ and,
2.  if any elements $g \in G$ can be written uniquely in the form $g = s_1^{\epsilon_1} \cdots s_n^{\epsilon_n}$ for some $\epsilon_i \in \mathbb{Z}$ by the relation $R$.

With the definition of an equivariant map in place ( A.2), disentangling a latent space relies on finding a corresponding group action $\cdot : G \times Z \to Z$ so that the symmetry structure of $W$ is reflected in an agent's representations, $Z$. This can be achieved if the following condition is satisfied:

$$g \cdot f(\boldsymbol{w}) = f(g \cdot \boldsymbol{w}) \quad \forall g \in G, \boldsymbol{w} \in W. \tag{2}$$

where $f : W \to Z$ is a mapping from world-states to an agent's representations. However, in general, one cannot control the nature of the generative process $b : W \to O$ leading from world-states to observations, $O$. In addition, without loss of generality, we can easily assume that the generative process $b$ is an equivariant map.

**Theorem 3.1.** *Suppose a generative process $b$ is an equivariant map satisfying , $g \cdot b(\mathbf{w}) = b(g \cdot \mathbf{w})$ $\forall g \in G, \mathbf{w} \in W$. Then, there exists a function $f$ that satisfies (2) if an inference process $h : O \to Z$ is an equivariant map satisfying,*

$$g \cdot h(\mathbf{o}) = h(g \cdot \mathbf{o}) \quad \forall g \in G, \mathbf{o} \in O. \tag{3}$$

*Proof.* Proof in the Appendix A.3. □

Following the Theorem 3.1, this assumption leads to the fact that the goal of disentangling is the same as finding an inference process $h : O \to Z$ satisfying,

$$g \cdot h(\boldsymbol{o}) = h(g \cdot \boldsymbol{o}) \quad \forall g \in G, \boldsymbol{o} \in O. \tag{4}$$

Although there is no guarantee that one can find a compatible action $\cdot : G \times Z \to Z$ satisfying (4), if $h$ is bijective then (4) can be expressed as follows,

$$g \cdot \boldsymbol{z} = h(g \cdot h^{-1}(\boldsymbol{z})) \tag{5}$$

However, if $h$ is a bijective function, simple neural network-based models cannot learn the overall equivariant map. Yet, the equivariant map, such as one outlined in equation 5 can be learned by autoencoders with inductive biases both on the model and the datasets, which is the central contribution of this paper. To show this mapping, let $h$ and $h^{-1}$ be an encoder, $E_\phi$, and a decoder, $D_\theta$, of an autoencoder. Then, the group action $\cdot : G \times Z \to Z$ can be defined as follows:

$$G \times Z \xrightarrow{id_G \times D_\theta} G \times O \xrightarrow{\cdot_o} O \xrightarrow{E_\phi} Z$$

This shows that the equivariant map can, indeed, be learned by an autoencoder. However, this is not without a number of challenges, which we discuss in Section 3.2 below.

## 3.2 Introducing Disentangled Representations into AutoEncoders

In deriving a disentangled representation, it is worth noting that, a vector addition, a basic and natural transformation, in the latent space enables natural transition between latent variables and nth root of unity is a cyclic group. We achieve a disentangled group action by the relations $R$ in the latent space by a map $(s_1^{\epsilon_1}, ..., s_n^{\epsilon_n})$ to $(e^{i\alpha_1\epsilon_1}, ..., e^{i\alpha_n\epsilon_n})$ and $W_i$ to $Z_i$ for some $\alpha_i \in (0,1)$. However complex numbers are undesirable in machine learning and so this is achieved by introducing an Euler encoding, $E$ on $\mathbb{R}^n$, into the AE architecture. We defined $E$ as follows:

$$E(\boldsymbol{z}) = (cos(2\pi z_1), sin(2\pi z_1), cos(2\pi z_2), sin(2\pi z_2), ..., cos(2\pi z_n), sin(2\pi z_n)) \tag{6}$$

where $n$ is the number of dimensions of the latent space.

**Theorem 3.2.** *Let $E$ be a Euler encoding and $\boldsymbol{A} : \mathbb{R}^{2n} \to \mathbb{R}^m$ be an injective linear transformation where $m > 2n$. For $\alpha \in (0,1)$ and $i \in \{1, ..., n\}$, let $T_i^\alpha : \mathbb{R}^n \to \mathbb{R}^n$ by $T_i^\alpha(\boldsymbol{x}) = (x_1, ..., x_i + \alpha, ..., x_n)$. Then $\boldsymbol{A} \cdot E(T_i^\alpha(\boldsymbol{z})) = \boldsymbol{A} \cdot E(T_j^\beta(\boldsymbol{z}))$ if and only if $i = j$ and $\alpha = \beta$.*

*Proof.* For $\boldsymbol{z} \in \mathbb{R}^n$, let $\boldsymbol{A} \cdot E(T_i^\alpha(\boldsymbol{z})) - \boldsymbol{A} \cdot E(T_j^\beta(\boldsymbol{z})) = \boldsymbol{0}$ and define

$$S_i^\alpha = \begin{bmatrix} \boldsymbol{I}_{2(i-1)} & & & \\ & cos(2\pi\alpha) & -sin(2\pi\alpha) & \\ & sin(2\pi\alpha) & cos(2\pi\alpha) & \\ & & & \boldsymbol{I}_{2(n-i)} \end{bmatrix}$$

Since $E(T_i^\alpha(\boldsymbol{z})) = S_i^\alpha \cdot E(\boldsymbol{z})$, $\boldsymbol{A} \cdot (S_i^\alpha - S_j^\beta) \cdot E(\boldsymbol{z}) = \boldsymbol{0}$.

(a) If $i \neq j$, then $\boldsymbol{A}$ is a zero transformation, which is a contradiction.

(b) If $i = j$, then $\alpha = \beta \pm k$, where $k \in \mathbb{Z}$, hence, $\alpha = \beta$.

$\square$

Since $E(T_i^\alpha(\boldsymbol{z})) = S_i^\alpha \cdot E(\boldsymbol{z})$ and $S_i^\alpha$ is an orthogonal transformation, the Euler encoding after translation on $Z$ can be considered as an orthogonal transformation of $E(\boldsymbol{z})$, which enables the changes of output from changes of different latent dimensions to be orthogonal. Nevertheless, there are still a number of practical challenges to overcome. These are: **(a) Number of Elements in a Subgroup**: The number of possible elements in the subgroups $N_j$ ($j = 1, \ldots, n$), or at least the relative ratio of the number of elements between the subgroups are not known a priori. This has a crucial role in introducing inductive biases on datasets, **(b) Robustness to Small Perturbations**: Because the proposed approach for disentanglement is deterministic, the model is not resilient to small perturbations (e.g., noise) (Camuto et al., 2021), which is essential for the model to behave in a robust manner when presented with unseen examples, and **(c) Spatial Distribution of Features**: An ideal factorized latent space must have the features spatially distributed in an equally likely manner. However, the equivariant map we discussed above alone may not be sufficient to address this issue. Although it is possible to address some of these concerns from a theoretical stand point, nearly all of these are addressable by carefully designing an architecture that exploits both the AE and the equivariant map principle discussed above, achieving the best possible disentanglement. We discuss these details in Section 3.3 below.

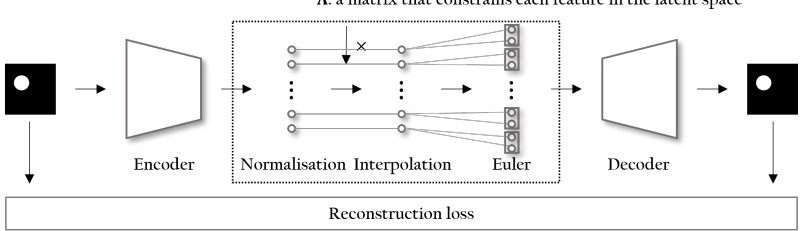

Figure 2: Illustration of the DAE architecture. The model includes the Euler encoding, and the outputs from the interpolation layer are mapped to cosine and sine values.

## 3.3 ARCHITECTURE OF THE DAE

In mapping our theory to an architecture, we build on the AE model, which constitutes an encoder, that maps the observation space $O$ to a factorized latent space $Z'$, followed by the disentangling process that factorizes/disentangles the latent space $Z$ to $Z'$, and finally the decoding layer, that maps the factorized latent $Z'$ to the regenerated observation space $O$. Each of the concerns that were discussed in Section 3.2 (a) through (c) are handled by a network of layers in our architecture. We show this model in Figure 2 and describe how this model addresses the relevant concerns next.

**(a) Number of Elements in a Subgroup:** Although the number of elements in a subgroup is not known a priori, the number or the relative ratio of the possible number of elements across subgroups can be estimated using techniques that can extract the variance information from compressed information, such as principal component analysis (PCA) (Jolliffe, 2002), independent component analysis (ICA) (Hyvärinen & Oja, 2000), or even through neural networks (Kingma & Welling, 2013; Burgess et al., 2018; Mondal et al., 2021). In this paper, for the reasons of simplification, we will be using the PCA technique. Since the singular values from PCA are proportional to the variances of the principal components of compressed data, these values are used to obtain a relative ratio of the number of possible element in the subgroups (Wall et al., 2003). Then, all singular values are divided by the maximum values, and rounded to one decimal places, and then values smaller than 1 are replaced with hyperparameter $\alpha$. The relevant algorithm is shown in Algorithm 1 in the Appendix. Let $\Lambda$ be the relative ratio of the possible number of elements across subgroups obtained from Algorithm 1.

**(b) Uniform Spatial Distribution of Features:** To ensure that each feature is equally/likely distributed across the latent space and falls within the $(0, 1)$ (which lets $\boldsymbol{A} \cdot E(T_i^\alpha(\boldsymbol{z})) = \boldsymbol{A} \cdot E(T_j^\beta(\boldsymbol{z}))$ if and only if $i = j$ and $\alpha = \beta$ in Theorem 3.2), we introduce a normalization layer, where we apply batch min-max normalization to the outputs of the encoder. As minimum and maximum values vary from batch (mini-batch) to batch (mini-batch), we update the moving minimum and maximum values during the training process, and use them during the test phase, akin to a batch normalization layer (Ioffe & Szegedy, 2015). The minimum and maximum values are also initialized close to the middle point of $[0, 1)$ to facilitate learning. This is then followed by scaling by $\Lambda$ to account for different number of possible elements for different features.

**(c) Robustness to Small Perturbations:** Robustness is achieved by introducing an Interpolation layer that performs Gaussian interpolation on the output of the normalized latent space, following Vincent et al. (2010); Berthelot et al. (2018). Gaussian interpolation is used to map unseen examples to known examples, and to make the latent space locally smooth. However, since the proposed model is deterministic, it is important to map a number of unseen examples to the learned representations. This is achieved by adding a weight-sensitive Gaussian noise to the outputs of the previous layer during training, which is obtained based on the closest proximal distance of each dimension of the representations. The relevant algorithm is shown in Algorithm 2 in the Appendix. It is worth noting that this layer will not be used during the inference phase.

## 3.4 A NOVEL METRIC FOR QUANTIFYING DISENTANGLEMENT: GF-SCORE

Nearly all of the existing set of metrics outlined in the literature for quantifying linear symmetry-based disentanglement require ground truth labels. Here, we propose a new metric, namely, Grid

Fitting Score (GF-Score), to achieve the same purpose without the need for labels. Our hypothesis is that performing independent disentangled actions on a symmetry group causes the corresponding subspace to form a grid-shape latent space. This can be exploited by generating a square grid to include latent variables, and by measuring the mean of the minimum distances from the square grid to latent variables to signify the quality of disentanglement. If the latent variables fit perfectly into the square grid, it would imply that the model achieves perfect linear disentanglement, and we can mark this as zero score. Therefore, the lower the GF-Score is, the better the disentanglement is. The relevant algorithm is shown in Appendix A 3.

## 4 EVALUATION AND RESULTS

### 4.1 EVALUATION METHOD

Our evaluation involves comparing the performance of the proposed approach against ten baseline models across six datasets using six disentanglement metrics. The code will be publicly available when the paper is published. We outline these details below.

**Datasets:** One of the critical challenges around evaluating disentanglement is identifying suitable datasets. It is difficult to identify common datasets to study this problem. In the literature, different datasets have been used for different purposes. For example, dSprite (Matthey et al., 2017), 3D Chair (Burgess & Kim, 2018) and CelebA (Liu et al., 2015) datasets have been used in $\beta$-VAE, $\beta$-TCVAE, and FVAE. Although these datasets are useful to understand the traversal order of the latent space, they lack a clear underlying group structure. As such, here, we utilize the datasets that have been first utilized in Higgins et al. (2018), with relevant enhancements, which we describe in the Appendix (See A.10). In addition to this dataset, we use five more datasets containing clear underlying group structure, namely, 2D Arrow, 3D Airplane (Tonnaer et al., 2022), 3D Teapots (East-wood & Williams, 2018), 3D Shape (Burgess & Kim, 2018) and 3D Face Model (Paysan et al., 2009) datasets. Finally, to demonstrate the performance on complex datasets, we use the Blood Cell Acevedo et al. (2020) and the Sprites Reed et al. (2015) datasets (see results in the Appendix).

**Baseline Models:** We considered ten different baselines models for our evaluation, namely, plain AE, vanilla VAE, $\beta$-VAE, $\beta$-TCVAE, CCI-VAE, FVAE, InfoVAE, DIPVAE, WAE and LSBDVAE. For DIPVAE, we only test DIPVAE-I, owing to the reasons of that DIPVAE-II model works better only for cases where the dimension of the latent space is larger than the ground truth factors, which is not the case for us. Furthermore, as the proposed technique is purely an AE-based method, we have not included any GAN-specific baselines. To render a fair evaluation mechanism, we used the same encoder and decoder architectures, and same latent space dimensions (for each baseline model), which are used in Higgins et al. (2017); Kim & Mnih (2018); Quessard et al. (2020); Tonnaer et al. (2022) throughout the evaluation. Please see the Appendix (See A.8) for additional details.

**Performance Metrics:** As discussed in Section 2.1, a large number of metrics can be used to study the performance of disentanglement, depending on the nature of the dataset, access to ground truth, availability of latent factors, and the number of dimensions in the latent space. We use two types of metrics: **(a) Visualization of the latent space**, and **(b) Numerical disentanglement score**. The former permits one to visualize the orthogonality between features, and can be used to demonstrate how the latent traversal is achieved by the model and grid structures in the latent space. The second approach provides a quantifiable method for assessing the disentanglement. Collectively, we have used six metrics, including five supervised metrics accounting for each of the disentanglement metric classes (see Section 2.1), namely, **z-diff** and **z-min** from the intervention-based, **dci-rf** from the predictor-based, and **jemmig** and **dcimig** from the information-based metric classes in order to measure disentanglement, completeness and informativeness, along with GF-Score (See 3.4). In Locatello et al. (2019), it was shown that variances of all metrics are large with random seeds and it disturbs the comparison between different models. Hence, we run all the models on each data set for 20 different random seeds and select the random seed with the highest total score over these metrics.

### 4.2 RESULTS AND DISCUSSIONS

Our evaluation has produced a considerable volume of results, and for the reasons of brevity, we present two sets of results here, namely, (i) we show the reconstructions of latent traversals for the 2D Arrow, 3D Airplane, $XYCS$, 3D shape, 3D Teapots and 3D Face Model datasets in Figure 3,

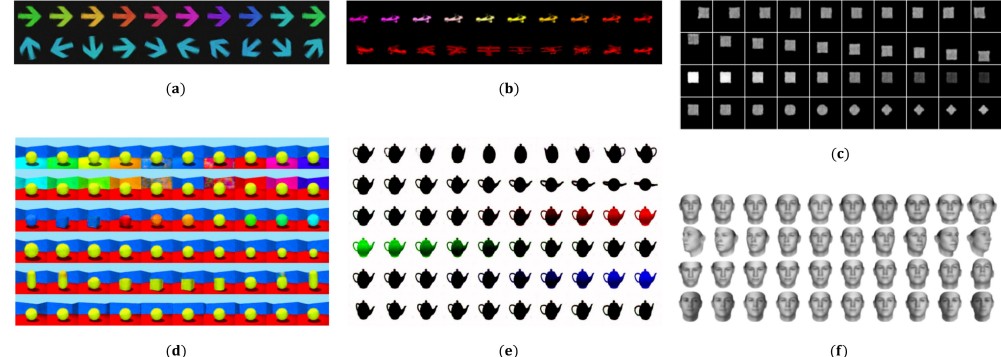

Figure 3: Reconstructions of latent traversals across each latent dimension obtained by the DAE for the (a) 2D Arrow (color and shape), (b) 3D Airplane (color and shape), (c) $XYCS$ ($x$ position, $y$ position, color and shape), (d) 3D Shape (floor hue, wall hue, object hue, scale, shape and orientation), (e) 3D Teapots (azimuth, elevation, red, green, blue and extra) and (f) 3D Face Model datasets (face id, azimuth, elevation and lighting).

Table 1: Differences in scores between $XY/XYCS$ datasets. Absolute and percentage of change from $XY$ to $XYCS$ are shown. Percentage changes closer to the zero are desirable.

| Models | Disentanglement scores | | | | |
|---|---|---|---|---|---|
| | z-diff ↑ | z-var ↑ | dci-rf ↑ | jemmig ↑ | dcimig ↑ |
| **DAE** | $^{1.00}/_{1.00}$ (0.0%) | $^{1.00}/_{1.00}$ (0.0%) | $^{0.99}/_{0.94}$ (-5.0%) | $^{0.81}/_{0.67}$ (-17.2%) | $^{0.80}/_{0.75}$ (-6.2%) |
| $\beta$-VAE | $^{1.00}/_{0.79}$ (-21.0%) | $^{1.00}/_{0.46}$ (-54.0%) | $^{0.91}/_{0.08}$ (-91.2%) | $^{0.65}/_{0.18}$ (-72.3%) | $^{0.63}/_{0.13}$ (-79.3%) |
| $\beta$-TCVAE | $^{1.00}/_{0.72}$ (-28.0%) | $^{1.00}/_{0.41}$ (-59.0%) | $^{0.93}/_{0.15}$ (-83.8%) | $^{0.70}/_{0.24}$ (-65.7%) | $^{0.69}/_{0.14}$ (-79.7%) |
| CCI-VAE | $^{1.00}/_{1.00}$ (0.0%) | $^{1.00}/_{1.00}$ (0.0%) | $^{0.98}/_{0.63}$ (-35.0%) | $^{0.78}/_{0.47}$ (-39.7%) | $^{0.76}/_{0.46}$ (-39.4%) |
| FVAE | $^{1.00}/_{1.00}$ (0.0%) | $^{1.00}/_{0.91}$ (-9.0%) | $^{0.96}/_{0.18}$ (-80.8%) | $^{0.73}/_{0.21}$ (-69.5%) | $^{0.70}/_{0.13}$ (-80.5%) |
| InfoVAE | $^{1.00}/_{0.92}$ (-8.0%) | $^{1.00}/_{0.54}$ (-46.0%) | $^{0.90}/_{0.21}$ (-76.6%) | $^{0.64}/_{0.27}$ (-57.8%) | $^{0.67}/_{0.13}$ (-76.6%) |
| DIPVAE | $^{1.00}/_{1.00}$ (0.0%) | $^{1.00}/_{0.44}$ (-56.0%) | $^{0.98}/_{0.32}$ (-67.3%) | $^{0.78}/_{0.28}$ (-64.1%) | $^{0.78}/_{0.11}$ (-85.8%) |
| LSBDVAE | $^{1.00}/_{1.00}$ (0.0%) | $^{1.00}/_{0.76}$ (-24.0%) | $^{0.96}/_{0.38}$ (-60.4%) | $^{0.72}/_{0.30}$ (-58.3%) | $^{0.70}/_{0.28}$ (-60.0%) |

along with the percentage of changes when color and shape features are added to $XY$ dataset in Table 1, and, (ii) we present the disentanglement scores of top two performing models across all datasets for all metrics in Tables 1 and 2. We provide the remaining set of results (reconstructions of latent traversals and disentanglement scores of all models), and other relevant details (such as hyper-parameters, and network architectures A.7 and A.8) as part of the Appendix.

### 4.2.1 LATENT SPACE VISUALIZATION

We show the disentangled (2D) latent space for the $XY$, 2D Arrow and 3D Airplane datasets, which have two underlying factors, in Figure 1 (also see Table 14 in Appendix A for details of relevant hyperparameters). As can be seen in the Figure 1, the proposed model, in general, provides the ideal grid-shape outlined in Higgins et al. (2018). The plain AE, vanilla VAE, Info-VAE and WAE models offer the worst performance. Other models, such as $\beta$-VAE, $\beta$-TCVAE, CCI-VAE, and DIPVAE models also come closer to the ideal pattern in the three datasets, and thus most models are able to disentangle $x$ and $y$ positions in $XY$ datasets, and rotation and color factors in 2D Arrow and 3D Airplane datasets. However, when color or shape feature is added to the $XY$ dataset (i.e., for $XYC$, $XYS$ and $XYCS$ datasets), the disentanglement can become a significant challenge, other than for the proposed model (See Figure 11 in Appendix). In addition to these, pairs of latent spaces, and reconstructions of latent traversals (across each latent dimension) of six datasets for the DAE are shown in Figure 3). (Also the Appendix A.11 for more results.)

### 4.2.2 DISENTANGLEMENT SCORES

We present the supervised disentanglement scores and their percentage changes when color and shape features are added to the $XY$ dataset in Table 1, with the changes measured relative to the

Table 2: Disentanglement scores for the 2D Arrow, 3D Airplane, 3D Teapots, 3D Shape and 3D Face Model datasets

| Datasets | 2D Arrow | | 3D Airplane | | 3D Teapots | | 3D Shape | | 3D Face Model | |
|---|---|---|---|---|---|---|---|---|---|---|
| Metrics/Models | **DAE** | DIPVAE | **DAE** | DIPVAE | **DAE** | DIPVAE | **FVAE** | DAE | **DAE** | $\beta$-TCVAE |
| z-diff $\uparrow$ | 1.00 | 1.00 | 1.00 | 1.00 | 1.00 | 1.00 | 1.00 | 1.00 | 1.00 | 1.00 |
| z-var $\uparrow$ | 0.85 | 0.96 | 1.00 | 0.96 | 1.00 | 1.00 | 1.00 | 0.93 | 1.00 | 0.92 |
| dci-rf $\uparrow$ | 0.88 | 0.85 | 0.80 | 0.54 | 0.89 | 0.84 | 0.99 | 0.99 | 0.65 | 0.59 |
| jemmig $\uparrow$ | 0.80 | 0.75 | 0.79 | 0.51 | 0.54 | 0.52 | 0.86 | 0.87 | 0.48 | 0.53 |
| dcimig $\uparrow$ | 0.79 | 0.72 | 0.75 | 0.43 | 0.53 | 0.50 | 0.88 | 0.90 | 0.47 | 0.47 |
| GF ($e^{-100}$) $\downarrow$ | 0.30 | 2.55 | 0.19 | 7.66 | 0.002 | 0.26 | 0.20 | 0.0009 | 0.02 | 0.37 |

$XY$ dataset, and it is worth noting that the added features have smaller variances than $x$ and $y$ positions. As can be seen, in general, nearly, all models suffer from the performance drop except a few. The CCI-VAE is the only model that performs as good as the proposed model for the **z-diff** and **z-min** metrics. The proposed model shows the smallest drop across three remaining metrics, namely, **dci-rf**, **jemmig** and **dcimig**. While the largest drop in the proposed model is $17.2\%$, the scores fall by between $35.0\%$ and $91.2\%$ for the other models.

The disentanglement scores for the top two performing models for all datasets (except the $XYS$ dataset) are shown in Tables 1 and 2, with the best performing model highlighted in boldface. From these results (including those in the appendix), the following observations can be drawn: Firstly, the proposed model outperforms all models across all metrics for the 2D Toy (covering $XY$, $XYC$, $XYS$, $XYCS$ datasets), 3D Airplane and 3D Teapots datasets (See Table 8 and Table 7). Secondly, for the remaining five datasets, DAE offered the best score across four of those datasets (2D Arrow, 3D Airplane, 3D Teapots and 3D Face Model) while offering the second best performance for the 3D Shape dataset. Where the DAE offered the second best performance, DAE still achieve higher scores **jemmig** and **dcimig** and the same scores in **z-diff** and **z-min**. Thirdly, for the 2D Toy dataset, the proposed model maintains the reconstruction loss as small as possible whilst offering improved disentanglement scores (See Figures 25-28). On the other hand, the reconstruction losses for the $\beta$-VAE, $\beta$-TCVAE and CCI-VAE models increase along with their disentanglement scores. Finally, GF-Score shows that the proposed model perfectly fits the latent space into a grid structure across all datasets and baseline models. Based on the GF-Score from Table 4 to 10, a model without regularizer, such as AE, fails to form a grid structure in the latent space.

## 5 CONCLUSIONS

In the context of representation learning, being able to factorize or disentangle the latent space dimensions is crucial for obtaining latent representations that are composed of multiple, independent factors of variations. Literature around disentanglement methods are either predominantly supervised or semi-supervised, and as such, either labels or pairs of images are required or achieved via factorizing the aggregated posterior in the latent space.

In this paper, we presented a non-probabilistic, deterministic model, namely, disentangling autoencoder or DAE, addressing a number of issues found in the literature. We also demonstrated how to realize the disentanglement conceptualized in Higgins et al. (2018) for the first time, especially without requiring labels or pairs of images. Our approach exploits the Euler encoding that makes the subspaces of the latent space independent of one another. Along with the architectural details, we also presented a novel metric for quantifying disentanglement, namely, GF-Score. Our detailed evaluations, performed against a large number of AE-based models, using considerably a large number of metrics show that our model can easily offer superior disentanglement performance when compared against a number of existing methods across a number of datasets.

Although the results are encouraging, a number of aspects remain to investigated, including, evaluation of the proposed model on datasets that lack underlying group structure, understanding the effect of the choice of the latent dimension on the outcomes, and to evaluate different latent space smoothing algorithms, to mention a few.

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

# A APPENDIX

## A.1 A REVIEW OF GROUP THEORY

More details of definitions and theorems can be found in Dummit & Foote (2004).

**Definition A.1.** *A group is an ordered pair* $(G, \star)$ *where* $G$ *is a set and* $\star$ *is a binary operation on* $G$ *satisfying the following axioms:*

1. *Associativity:* $(a \star b) \star c = a \star (b \star c)$, *for all* $a, b, c \in G$,

2. *Identity: there exists an element* $e$ *in* $G$, *called an identity of* $G$, *such that for all* $a \in G$, $a \star e = e \star a$,

3. *Inverses: for each* $a \in G$ *there is an element* $a^{-1}$ *of* $G$, *called an inverse of* $a$, *such that* $a \star a^{-1} = a^{-1} \star a = e$.

*We shall write the operation* $a \star b$ *as* $ab$.

**Definition A.2.** *A group action of a group* $G$ *on a set* $A$ *is a map* $\cdot : G \times A \to A$ *by* $\cdot(g, a) = g \cdot a$ *satisfying the following properties:*

1. $g_1 \cdot (g_2 \cdot a) = (g_1 g_2) \cdot a$, *for all* $g_1, g_2 \in G$, $a \in A$, *and*

2. $e \cdot a = a$, *for all* $a \in A$.

**Definition A.3.** *Let* $G$ *be a group. The subset* $H$ *of* $G$ *is a subgroup of* $G$ *if* $H$ *is nonempty and* $x, y \in H$ *implies* $x^{-1} \in H$ *and* $xy \in H$.

**Definition A.4.** *A group* $H$ *is cylic if* $H$ *can be generated by a single element, i.e., there is some element* $x \in H$ *such that* $H = \{x^n | n \in Z\}$, *and will be denoted by* $< x >$.

## A.2 DISENTANGLED REPRESENTATION

The notion of a disentangled representation is mathematically defined using the concept of symmetry in Higgins et al. (2018). For example, horizontal and vertical translations are symmetry transformations in two-dimensional grid, and, hence, such transformations change the location of an object in this two-dimensional grid. From the definitions of a symmetry group in Higgins et al. (2018), a symmetry group can be decomposed as a product of multiple subgroups, if suitable subgroups can be identified. This can render an intuitive method to disentangle the latent space, if subgroups that independently act on subspaces of a latent space, can be found. If actions by transformations of each subgroup only affect the corresponding subspace, the actions are called *disentangled group actions*. In other words, disentangled group actions only change a specific property of the state of an object, and leaves the other properties invariant. If there is a transformation in a vector space of representations, corresponding to a disentangled group action, the representation is called a *disentangled representation*. We reproduce the formal definitions of disentangled group action and disentangled representation from Higgins et al. (2018), as Definitions A.5 and A.6, respectively.

**Definition A.5.** *Suppose that we have a group action* $\cdot : G \times X \to X$, *and the group* $G$ *decomposes as a direct product* $G = G_1 \times \cdots \times G_n$. *Let the action of the full group, and the actions of each subgroups be referred to as* $\cdot$ *and* $\cdot_i$, *respectively. Then, the action is disentangled if there is a decomposition* $X = X_1 \times \cdots \times X_n$, *and actions* $\cdot_i : G_i \times X_i \to X_i$, $i \in \{1, \cdots, n\}$ *such that:*

$$(g_1, \cdots, g_n) \cdot (\mathbf{x_1}, \cdots, \mathbf{x_n}) = (g_1 \cdot \mathbf{x_1}, \cdots, g_n \cdot \mathbf{x_n}) \tag{7}$$

*for all* $g_i \in G_i$ *and* $\mathbf{x_i} \in X_i$.

Now, to derive the definition of a disentangled representation from the definition of disentangled group action, consider a set of world-states, denoted by $W$. Furthermore, assume that: (a) there is a generative process $b : W \to O$ leading from world-states to observations, $O$, (b) and an inference process $h : O \to Z$ leading from observations to an agent's representations, $Z$. With these, consider the composition $f : W \to Z$, $f = h \circ b$. In terms of transformation, assume that these transformations are represented by a group $G$ of symmetries acting on $W$ via an action $\cdot : G \times W \to W$.

The overarching goal of disentangling the latent space now relies on finding a corresponding action $\cdot : G \times Z \to Z$ so that the symmetry structure of $W$ is reflected in $Z$. In other words, an action on $Z$ corresponding to the action on $W$ is desirable. This can be achieved if the following condition is satisfied:

$$g \cdot f(\mathbf{w}) = f(g \cdot \mathbf{w}) \quad \forall g \in G, \mathbf{w} \in W. \tag{8}$$

In other words, the action, $\cdot$, should commute with $f$, which adheres to the definition of the equivariant map, and thus, $f$ is an equivariant map, as shown below.

$$
\begin{array}{ccc}
G \times W & \xrightarrow{\ \cdot \mathbf{w}\ } & W \\
\downarrow{\scriptstyle id_G \times f} & & \downarrow{\scriptstyle f} \\
G \times Z & \dashrightarrow{\ \cdot \mathbf{z}\ } & Z
\end{array}
$$

From Higgins et al. (2018), a disentangled representation can be defined as follows:

**Definition A.6.** *The representation $Z$ is disentangled with respect to $G = G_1 \times \cdots \times G_n$ if*

1. *There is an action $\cdot : G \times Z \to Z$,*

2. *The map $f : W \to Z$ is equivariant between the actions on $W$ and $Z$, and*

3. *There is a decomposition $Z = Z_1 \times \cdots \times Z_n$ or $Z = Z_1 \oplus \cdots \oplus Z_n$ such that each $Z_i$ is fixed by the action of all $G_j$, $j \neq i$ and affected only by $G_i$.*

### A.3  PROOF

Proof of Theorem 3.1

*Proof.* Suppose that there $b$ is an equivariant map defined in Theorem 3.1 and $h$ is an equivariant map. Then

$$g \cdot f(\mathbf{w}) = g \cdot h(b(\mathbf{w})) \tag{9}$$
$$= h(g \cdot b(\mathbf{w})) \tag{10}$$
$$= h(b(g \cdot \mathbf{w})) \tag{11}$$
$$= f(g \cdot \mathbf{w})) \tag{12}$$
$$\tag{13}$$

$\forall g \in G, \mathbf{w} \in W.$ $\qquad\square$

### A.4  COMPARISON OF DIFFERENT VAE-BASED MODELS

In our evaluation, we compare the proposed model against seven other VAE-based derivatives, namely, vanilla VAE, $\beta$-VAE, $\beta$-TCVAE, CCI-VAE, FVAE, InfoVAE and WAE. All these models vary based on the underlying regularizer $L_{reg}(\boldsymbol{\phi})$. For example, the $\beta$-VAE model constraints on the latent space using $\beta$ to limit the capacity of the latent space, which encourages the model to learn the most efficient representation of the data. The regularization term of these different models (Column 2) are summarized in Table 3 along with relevant notes (Column 3).

Table 3: Comparison of different VAE-based models w.r.t the regularizers they employ.

| Model | $L_{reg}(\boldsymbol{\phi})$ | Notes |
|---|---|---|
| VAE | $D_{\mathrm{KL}}(q_{\boldsymbol{\phi}}(\boldsymbol{z}|\boldsymbol{x})\|p(\boldsymbol{z}))$ | — |
| $\beta$-VAE | $\beta D_{\mathrm{KL}}(q_{\boldsymbol{\phi}}(\boldsymbol{z}|\boldsymbol{x})\|p(\boldsymbol{z}))$ | Usually, $\beta$ is greater than 1 |
| $\beta$-TCVAE | $I(\boldsymbol{z},\boldsymbol{x}) + \beta D_{\mathrm{KL}}(q(\boldsymbol{z})\| \prod_j q(\boldsymbol{z}_j)) + \sum_j D_{\mathrm{KL}}(q(\boldsymbol{z}_j)\|p(\boldsymbol{z}_j))$ | $I(\cdot,\cdot)$ is a mutual information |
| CCI-VAE | $\beta|D_{\mathrm{KL}}(q_{\boldsymbol{\phi}}(\boldsymbol{z}|\boldsymbol{x})\|p(\boldsymbol{z})) - C|$ | $C$ is a capacity |
| FVAE | $D_{\mathrm{KL}}(q_{\boldsymbol{\phi}}(\boldsymbol{z}|\boldsymbol{x})\|p(\boldsymbol{z})) + \gamma D_{\mathrm{KL}}(q(\boldsymbol{z})\| \prod_j q(\boldsymbol{z}_j)))$ | - |
| InfoVAE | $D_{\mathrm{KL}}(q_{\boldsymbol{\phi}}(\boldsymbol{z}|\boldsymbol{x})\|p(\boldsymbol{z})) + \lambda MMD(q_{\boldsymbol{\phi}}(\boldsymbol{z}|\boldsymbol{x}), p(\boldsymbol{z}))$ | $MMD(\cdot,\cdot)$ is Maximum Mean Discrepancy |
| DIPVAE | $D_{\mathrm{KL}}(q_{\boldsymbol{\phi}}(\boldsymbol{z}|\boldsymbol{x})\|p(\boldsymbol{z})) + \lambda_{od} \sum_{i\neq j}[Cov_{p(\boldsymbol{x})}[\mu_{\boldsymbol{\phi}}(\boldsymbol{x})]]_{i,j}^2$ $+\lambda_d \sum_i ([Cov_{p(\boldsymbol{x})}[\mu_{\boldsymbol{\phi}}(\boldsymbol{x})]]_{i,i} - 1)^2$ | $Cov$ is a covariance and $\mu_{\boldsymbol{\phi}}(\boldsymbol{x})$ is the output of an encoder. We set $\lambda_d = 10\lambda_{od}$ as suggested in Kumar et al. (2017) |
| WAE | $\lambda MMD(q_{\boldsymbol{\phi}}(\boldsymbol{z}|\boldsymbol{x}), p(\boldsymbol{z}))$ | $\lambda$ is a regularization coefficient |
| LSBDVAE | $\Delta D_{\mathrm{KL}}(q_{\boldsymbol{\phi}}(\boldsymbol{z}|\boldsymbol{x})\|p(\boldsymbol{z})) + \lambda D_{LSBD}$ | $\Delta D_{\mathrm{KL}}$ is a KL divergence used in Diffusion VAE |

## A.5 ALGORITHMS

---

**Algorithm 1:** Obtaining $\Lambda$ using PCA

---

**Input:** $X$: the entire dataset and $\alpha$: hyperparameter less than 1
**Output:** $\Lambda = [w_1, w_2, \cdots, w_n]$
If $n > 2$:
    $S = [s_1, s_2, \cdots, s_n]$: singular values from PCA$(X)$
    $\bar{S} = [\bar{s_1}, \bar{s_2}, \cdots, \bar{s_n}] = S/max(S)$
    $\Lambda = [w_1, w_2, \cdots, w_n]$: round to one decimal place of $\bar{S}$ ($\bar{S}$ of each dataset is shown in
Table 12.)
    If there exists $i$ such that $w_i < 1$, then $w_i = \alpha$
Otherwise:
    $\Lambda = [1, \alpha]$

---

**Algorithm 2:** Interpolation layer

---

**Input:** A mini-batch: $B = \{\boldsymbol{x}_1, ..., \boldsymbol{x}_m\}$.
**Output:** $\{\boldsymbol{y}_1, ..., \boldsymbol{y}_m\}$
Let $\boldsymbol{x}_i = (x_i^k)_{k=1,\cdots,n}$.
for $k$ in $\{1, 2, ..., n\}$
    for $i$ in $\{1, 2, ..., n\} - \{k\}$
        Denote weight $w_i^k = min_{j\in\{1,\cdots,m\}} d(x_i^k, x_j^k)$
        $y_i^k = x_i^k + w_i^k * \varepsilon$ where $\varepsilon \sim \mathcal{N}(0,1)$

---

**Algorithm 3:** Grid fitting score method

---

**Input:** $\boldsymbol{Z} = [\boldsymbol{Z}_{:,1}, ..., \boldsymbol{Z}_{:,n}]$: a matrix consisting of all latent variables obtained by a model.
  Each row corresponds to one latent variable.
**Output:** $S$
for $i$ in $\{1, 2, ..., n-1\}$
    for $j$ in $\{i, i+1, ..., n\}$
        Denote $\boldsymbol{Z}^{i,j} = [\boldsymbol{Z}_{:,i}, \boldsymbol{Z}_{:,j}]$: a two dimensional subspace of $\boldsymbol{Z}$
        Create $\boldsymbol{G}^{i,j}$: a set of variables from a square grid that fits $\boldsymbol{Z}^{i,j}$
        Let $d^{i,j} = 0$
        for $k$ in range(len($\boldsymbol{G}^{i,j}$)):
            $d^{i,j} + = distance(\boldsymbol{G}^{i,j}_{k,:}, \boldsymbol{Z}^{i,j}_{k,:})$
        $S^{i,j} = d^{i,j}/k$
$S$: the average of $S^{i,j}$

---

## A.6 DISENTANGLEMENT SCORES

Table 4: Disentanglement scores for the $XY$ dataset

| Models/Metrics | z-diff ↑ | z-var ↑ | dci-rf ↑ | jemmig ↑ | dcimig ↑ | GF $(e^{-100})$ ↓ | $MSE$ ↓ |
|---|---|---|---|---|---|---|---|
| DAE | **1.00** | **1.00** | **0.99** | **0.81** | **0.80** | **0.21** | 0.006 |
| AE | **1.00** | **1.00** | 0.79 | 0.55 | 0.50 | 8.67 | 0.003 |
| VAE | **1.00** | **1.00** | 0.84 | 0.58 | 0.54 | 8.52 | 0.002 |
| $\beta$-VAE | **1.00** | **1.00** | 0.91 | 0.65 | 0.63 | 3.82 | 0.003 |
| $\beta$-TCVAE | **1.00** | **1.00** | 0.93 | 0.70 | 0.69 | 4.81 | 0.004 |
| CCI-VAE | **1.00** | **1.00** | 0.98 | 0.78 | 0.76 | 3.25 | 0.009 |
| FVAE | **1.00** | **1.00** | 0.96 | 0.73 | 0.70 | 5.58 | 0.003 |
| InfoVAE | **1.00** | **1.00** | 0.90 | 0.64 | 0.60 | 8.06 | 0.002 |
| DIPVAE | **1.00** | **1.00** | 0.98 | 0.78 | 0.78 | 2.64 | 0.003 |
| WAE | **1.00** | **1.00** | 0.44 | 0.35 | 0.27 | 5.82 | 0.003 |
| LSBDVAE | **1.00** | **1.00** | 0.96 | 0.72 | 0.70 | 5.47 | 0.003 |

Table 5: Disentanglement scores for the $XYCS$ dataset

| Models/Metrics | z-diff ↑ | z-var ↑ | dci-rf ↑ | jemmig ↑ | dcimig ↑ | GF $(e^{-100})$ ↓ | $MSE$ ↓ |
|---|---|---|---|---|---|---|---|
| DAE | **1.00** | **1.00** | **0.94** | **0.67** | **0.75** | **0.01** | 0.002 |
| AE | 0.95 | 0.55 | 0.24 | 0.22 | 0.13 | 18.37 | 0.001 |
| VAE | 0.82 | 0.30 | 0.09 | 0.21 | 0.08 | 0.58 | 0.001 |
| $\beta$-VAE | 0.79 | 0.46 | 0.08 | 0.18 | 0.13 | 0.44 | 0.001 |
| $\beta$-TCVAE | 0.72 | 0.41 | 0.15 | 0.24 | 0.14 | 0.61 | 0.002 |
| CCI-VAE | **1.00** | **1.00** | 0.63 | 0.47 | 0.46 | 0.25 | 0.002 |
| FVAE | **1.00** | 0.91 | 0.18 | 0.21 | 0.13 | 0.10 | 0.004 |
| InfoVAE | 0.92 | 0.54 | 0.21 | 0.27 | 0.14 | 0.44 | 0.001 |
| DIPVAE | **1.00** | 0.44 | 0.32 | 0.28 | 0.11 | 0.26 | 0.001 |
| WAE | 0.93 | 0.38 | 0.06 | 0.15 | 0.05 | 0.42 | 0.003 |
| LSBDVAE | **1.00** | 0.76 | 0.38 | 0.30 | 0.28 | 0.54 | 0.001 |

Table 6: Disentanglement scores for the 2D Arrow dataset

| Models/Metrics | z-diff ↑ | z-var ↑ | dci-rf ↑ | jemmig ↑ | dcimig ↑ | GF $(e^{-100})$ ↓ | $MSE$ ↓ |
|---|---|---|---|---|---|---|---|
| DAE | **1.00** | 0.85 | **0.88** | **0.80** | **0.79** | **0.30** | 0.014 |
| AE | **1.00** | 0.30 | 0.00 | 0.29 | 0.17 | 4480.97 | 0.012 |
| VAE | 0.99 | 0.85 | 0.01 | 0.19 | 0.05 | 16.36 | 0.059 |
| $\beta$-VAE | **1.00** | 0.88 | 0.34 | 0.62 | 0.55 | 7.71 | 0.034 |
| $\beta$-TCVAE | **1.00** | 0.87 | 0.83 | 0.77 | 0.73 | 5.48 | 0.011 |
| CCI-VAE | **1.00** | 0.84 | 0.79 | 0.75 | 0.70 | 3.07 | 0.017 |
| FVAE | **1.00** | 0.90 | 0.46 | 0.58 | 0.52 | 17.09 | 0.022 |
| InfoVAE | **1.00** | 0.90 | 0.05 | 0.33 | 0.22 | 16.75 | 0.029 |
| DIPVAE | **1.00** | **0.96** | 0.85 | 0.75 | 0.72 | 2.55 | 0.010 |
| WAE | **1.00** | 0.62 | 0.07 | 0.24 | 0.10 | 1510.34 | 0.006 |
| LSBDVAE | **1.00** | 0.83 | 0.83 | **0.80** | 0.76 | 4.07 | 0.008 |

Table 7: Disentanglement scores for the 3D Airplane dataset

| Models/Metrics | z-diff $\uparrow$ | z-var $\uparrow$ | dci-rf $\uparrow$ | jemmig $\uparrow$ | dcimig $\uparrow$ | GF $(e^{-100})\downarrow$ | $MSE\downarrow$ |
|---|---|---|---|---|---|---|---|
| DAE | **1.00** | **1.00** | **0.80** | **0.79** | **0.75** | **0.28** | 0.011 |
| AE | **1.00** | .038 | 0.00 | 0.22 | 0.14 | 2505.92 | 0.003 |
| VAE | **1.00** | **1.00** | 0.01 | 0.36 | 0.28 | 14.96 | 0.013 |
| $\beta$-VAE | **1.00** | **1.00** | 0.09 | 0.44 | 0.35 | 7.92 | 0.015 |
| $\beta$-TCVAE | **1.00** | 0.96 | 0.28 | 0.52 | 0.44 | 15.11 | 0.009 |
| CCI-VAE | **1.00** | 0.95 | 0.13 | 0.49 | 0.41 | 5.60 | 0.013 |
| FVAE | **1.00** | **1.00** | 0.01 | 0.39 | 0.29 | 14.06 | 0.011 |
| InfoVAE | **1.00** | 0.98 | 0.17 | 0.48 | 0.37 | 11.86 | 0.014 |
| DIPVAE | **1.00** | 0.96 | 0.54 | 0.51 | 0.43 | 7.66 | 0.004 |
| WAE | **1.00** | 0.82 | 0.06 | 0.32 | 0.17 | 1392.81 | 0.002 |
| LSBDVAE | **1.00** | 0.94 | 0.18 | 0.49 | 0.41 | 11.90 | 0.002 |

Table 8: Disentanglement scores for the 3D Teapots dataset

| Models/Metrics | z-diff $\uparrow$ | z-var $\uparrow$ | dci-rf $\uparrow$ | jemmig $\uparrow$ | dcimig $\uparrow$ | GF $(e^{-100})\downarrow$ | $MSE\downarrow$ |
|---|---|---|---|---|---|---|---|
| DAE | **1.00** | **1.00** | **0.89** | **0.54** | **0.53** | **0.0031** | 0.002 |
| AE | 0.89 | 0.45 | 0.16 | 0.16 | 0.05 | 23.22 | 0.001 |
| VAE | **1.00** | 0.77 | 0.43 | 0.29 | 0.20 | 0.24 | 0.001 |
| $\beta$-VAE | 0.93 | 0.80 | 0.47 | 0.29 | 0.20 | 0.14 | 0.001 |
| $\beta$-TCVAE | **1.00** | 0.83 | 0.67 | 0.35 | 0.30 | 0.16 | 0.001 |
| CCI-VAE | 0.91 | 0.65 | 0.42 | 0.35 | 0.16 | 0.02 | 0.003 |
| FVAE | **1.00** | 0.79 | 0.50 | 0.32 | 0.25 | 0.17 | 0.001 |
| InfoVAE | **1.00** | 0.73 | 0.47 | 0.31 | 0.23 | 0.15 | 0.001 |
| DIPVAE | **1.00** | **1.00** | 0.84 | 0.52 | 0.50 | 0.13 | 0.001 |
| WAE | 0.78 | 0.52 | 0.15 | 0.15 | 0.04 | 0.20 | 0.001 |
| LSBDVAE | **1.00** | 0.68 | 0.51 | 0.33 | 0.25 | 0.15 | 0.001 |

Table 9: Disentanglement scores for the 3D Shape dataset

| Models/Metrics | z-diff $\uparrow$ | z-var $\uparrow$ | dci-rf $\uparrow$ | jemmig $\uparrow$ | dcimig $\uparrow$ | GF $(e^{-100})\downarrow$ | $MSE\downarrow$ |
|---|---|---|---|---|---|---|---|
| DAE | **1.00** | 0.93 | **0.99** | **0.87** | **0.90** | **0.0013** | 0.0008 |
| AE | 0.95 | 0.70 | 0.26 | 0.19 | 0.13 | 0.33 | 0.0006 |
| VAE | 0.96 | 0.81 | 0.31 | 0.27 | 0.22 | 0.14 | 0.0008 |
| $\beta$-VAE | **1.00** | 0.94 | 0.93 | 0.80 | 0.82 | 0.04 | 0.0022 |
| $\beta$-TCVAE | 0.89 | 0.75 | 0.79 | 0.77 | 0.74 | 0.02 | 0.0012 |
| CCI-VAE | 0.96 | 0.94 | 0.72 | 0.56 | 0.59 | 0.08 | 0.0008 |
| FVAE | **1.00** | **1.00** | **0.99** | 0.86 | 0.88 | 0.04 | 0.0007 |
| InfoVAE | 0.90 | 0.80 | 0.35 | 0.22 | 0.16 | 0.20 | 0.0007 |
| DIPVAE | 0.93 | 0.79 | 0.68 | 0.49 | 0.52 | 0.20 | 0.0007 |
| WAE | 0.74 | 0.45 | 0.09 | 0.17 | 0.08 | 0.14 | 0.0017 |
| LSBDVAE | **1.00** | 0.92 | 0.51 | 0.42 | 0.40 | 0.15 | 0.0006 |

Table 10: Disentanglement scores for the 3D Face Model dataset

| Models/Metrics | z-diff $\uparrow$ | z-var $\uparrow$ | dci-rf $\uparrow$ | jemmig $\uparrow$ | dcimig $\uparrow$ | GF $(e^{-100})\downarrow$ | $MSE\downarrow$ |
|---|---|---|---|---|---|---|---|
| DAE | **1.00** | **1.00** | **0.65** | 0.48 | **0.47** | **0.02** | 0.004 |
| AE | 0.69 | 0.47 | 0.10 | 0.19 | 0.05 | 100.60 | 0.003 |
| VAE | **1.00** | 0.69 | 0.47 | 0.34 | 0.20 | 0.47 | 0.002 |
| $\beta$-VAE | **1.00** | 0.92 | 0.60 | 0.52 | 0.44 | 0.34 | 0.005 |
| $\beta$-TCVAE | **1.00** | 0.92 | 0.59 | 0.53 | **0.47** | 0.37 | 0.005 |
| CCI-VAE | **1.00** | 0.90 | 0.61 | 0.53 | 0.44 | 0.31 | 0.002 |
| FVAE | **1.00** | 0.64 | 0.49 | 0.35 | 0.22 | 0.39 | 0.003 |
| InfoVAE | **1.00** | 0.90 | 0.57 | 0.50 | 0.38 | 0.44 | 0.002 |
| DIPVAE | **1.00** | 0.69 | 0.54 | 0.36 | 0.29 | 0.37 | 0.003 |
| WAE | **1.00** | 0.78 | 0.20 | 0.20 | 0.13 | 0.39 | 0.006 |
| LSBDVAE | **1.00** | 0.88 | 0.61 | **0.54** | 0.43 | 0.46 | 0.002 |

Table 11: Performance effects when removing the Euler layer (E) or the normalization layer (N)

| Datasets | $XY$ | | | $XYC$ | | | $XYS$ | | |
|---|---|---|---|---|---|---|---|---|---|
| Metrics/Models | DAE | w/o E | w/o N | DAE | w/o E | w/o N | DAE | w/o E | w/o N |
| z-diff ↑ | 1.00 | 1.00 | 1.00 | 1.00 | 1.00 | 1.00 | 1.00 | 1.00 | 1.00 |
| z-var ↑ | 1.00 | 1.00 | 1.00 | 1.00 | 0.68 | 0.58 | 1.00 | 0.80 | 0.36 |
| dci-rf ↑ | 0.99 | 0.59 | 0.95 | 0.99 | 0.44 | 0.26 | 0.99 | 0.54 | 0.11 |
| jemmig ↑ | 0.81 | 0.38 | 0.71 | 0.82 | 0.50 | 0.27 | 0.77 | 0.48 | 0.20 |
| dcimig ↑ | 0.80 | 0.33 | 0.70 | 0.83 | 0.63 | 0.27 | 0.80 | 0.61 | 0.19 |
| GF ($e^{-100}$) ↓ | 0.21 | 1.00 | 0.86 | 0.03 | 0.07 | 0.48 | 0.02 | 0.21 | 2.07 |

| Datasets | $XYCS$ | | | 2D Arrow | | | 3D Airplnae | | |
|---|---|---|---|---|---|---|---|---|---|
| Metrics/Models | DAE | w/o E | w/o N | DAE | w/o E | w/o N | DAE | w/o E | w/o N |
| z-diff ↑ | 1.00 | 0.99 | 0.82 | 1.00 | 1.00 | 1.00 | 1.00 | 1.00 | 1.00 |
| z-var ↑ | 1.00 | 0.82 | 0.17 | 0.85 | 0.65 | 0.57 | 1.00 | 1.00 | 0.40 |
| dci-rf ↑ | 0.94 | 0.57 | 0.07 | 0.88 | 0.04 | 0.00 | 0.80 | 0.00 | 0.00 |
| jemmig ↑ | 0.67 | 0.47 | 0.15 | 0.80 | 0.19 | 0.25 | 0.79 | 0.35 | 0.13 |
| dcimig ↑ | 0.75 | 0.59 | 0.06 | 0.79 | 0.53 | 0.19 | 0.75 | 0.27 | 0.12 |
| GF ($e^{-100}$) ↓ | 0.01 | 0.21 | 0.59 | 0.30 | 6.73 | 1213204.08 | 0.28 | 1.97 | 143594.36 |

| Datasets | 3D Teapots | | | 3D Shape | | | 3D Face Model | | |
|---|---|---|---|---|---|---|---|---|---|
| Metrics/Models | DAE | w/o E | w/o N | DAE | w/o E | w/o N | DAE | w/o E | w/o N |
| z-diff ↑ | 1.00 | 0.99 | 1.00 | 1.00 | 0.87 | 0.90 | 1.00 | 0.99 | 0.73 |
| z-var ↑ | 1.00 | 0.81 | 0.92 | 0.93 | 0.70 | 0.74 | 1.00 | 0.57 | 0.42 |
| dci-rf ↑ | 0.89 | 0.43 | 0.62 | 0.99 | 0.65 | 0.62 | 0.65 | 0.28 | 0.26 |
| jemmig ↑ | 0.54 | 0.22 | 0.40 | 0.87 | 0.53 | 0.50 | 0.48 | 0.31 | 0.22 |
| dcimig ↑ | 0.53 | 0.16 | 0.31 | 0.90 | 0.61 | 0.47 | 0.47 | 0.19 | 0.12 |
| GF ($e^{-100}$) ↓ | 0.0031 | 0.16 | 0.32 | 0.0013 | 0.47 | 0.44 | 0.02 | 0.32 | 0.71 |

## A.7 HYPERPARAMETERS

Table 12: $\bar{S}$ values for different datasets

| Dataset | $\bar{S}$ |
|---|---|
| $XY$ | [1.0, 1.0] |
| $XYC$ | [1.0, 1.0, 0.8] |
| $XYS$ | [1.0, 1.0, 0.8] |
| $XYCS$ | [1.0, 1.0, 0.8, 0.8] |
| 2D Arrow | [1.0, 1.0] |
| 3D Airplane | [1.0, 1.0] |
| 3D Teapots | [1.0, 0.8, 0.8, 0.4, 0.3, 0.3] |
| 3D Shape | [1.0, 1.0, 1.0, 1.0, 0.5, 0.5] |
| 3D Face Model | [1.0, 0.4, 0.4, 0.3] |

Table 13: All hyperparameters for models.

| Model | Values for $XYCS$ dataset | Extra values for the other datasets |
|---|---|---|
| DAE ($\alpha$) | $[1.0, 0.5, 0.1, 0.05, 0.01]$ | – |
| $\beta$-VAE ($\beta$) | $[2.0, 4.0, 8.0, 16.0, 32.0, 64.0, 128.0]$ | – |
| $\beta$-TCVAE ($\beta$) | $[2.0, 4.0, 8.0, 16.0, 32.0, 64.0, 128.0]$ | – |
| CCI-VAE (C) | $[100.0, 500., 1000.0]$ | [50.0] |
| FVAE ($\gamma$) | $[1.0, 5.0, 10.0, 20.0, 30.0, 40.0, 50.0]$ | – |
| InfoVAE ($\lambda$) | $[[100.0, 500., 1000.0]$ | [50.0, 2000.0] |
| DIPVAE ($\lambda$) | $[1.0, 2.0, 5.0, 10.0, 20.0, 50.0, 100.0]$ | – |
| WAE ($\lambda$) | $[1.0, 5.0, 10.0, 20.0, 30.0, 40.0, 50.0]$ | – |
| LSBD ($\lambda$) | $[1.0, 10.0, 100.0]$ | – |

Table 14: Best hyperparameters for models for the 2D datasets.

| Model / Dataset | $XY$ | $XYC$ | $XYS$ | $XYCS$ | 2D Arrow |
|---|---|---|---|---|---|
| DAE ($\alpha$) | 1.0 | 0.001 | 0.001 | 0.01 | 0.5 |
| $\beta$-VAE ($\beta$) | 16 | 32 | 64 | 16 | 16 |
| $\beta$-TCVAE ($\beta$) | 32 | 128 | 128 | 32 | 8 |
| CCI-VAE (C) | 500 | 100 | 100 | 100 | 50 |
| FVAE ($\gamma$) | 1 | 40 | 1 | 500 | 40 |
| InfoVAE ($\lambda$) | 100 | 20 | 1 | 500 | 50 |
| DIPVAE ($\lambda$) | 50 | 100 | 2 | 100 | 20 |
| WAE ($\lambda$) | 1 | 20 | 40 | 50 | 10 |
| LSBD ($\lambda$) | 100 | 1 | 1 | 1 | 10 |

Table 15: Best hyperparameters for models for the 3D datasets.

| Model / Dataset | 3D Airplane | 3D Teapots | 3D Shape | 3D Face Model |
|---|---|---|---|---|
| DAE ($\alpha$) | 0.05 | 0.01 | 0.01 | 0.05 |
| $\beta$-VAE ($\beta$) | 16 | 6 | 64 | 16 |
| $\beta$-TCVAE ($\beta$) | 16 | 6 | 32 | 32 |
| CCI-VAE (C) | 500 | 50 | 100 | 100 |
| FVAE ($\gamma$) | 20 | 1 | 5 | 1 |
| InfoVAE ($\lambda$) | 50 | 50 | 100 | 2000 |
| DIPVAE ($\lambda$) | 100 | 20 | 1 | 2 |
| WAE ($\lambda$) | 5 | 10 | 50 | 50 |
| LSBD ($\lambda$) | 10 | 100 | 10 | 10 |

## A.8 ENCODER AND DECODER ARCHITECTURES

Table 16: Architecture for the 2D Toy Dataset

| Encoder | Decoder |
|---|---|
| Input $84{\times}84{\times}1$ image | $3{\times}3$ 1 Conv $\downarrow$, Sigmoid |
| $10{\times}10$ 8 Conv $\downarrow$, BN, LReLU | $10{\times}10$ 1 Conv $\uparrow$, BN, LReLU |
| $10{\times}10$ 16 Conv $\downarrow$, BN, LReLU | $10{\times}10$ 8 Conv $\uparrow$, BN, LReLU |
| FC 64 | FC 256, LReLU |
| FC The number of features | FC 64, LReLU |

Table 17: Architecture for the 2D Arrow and 3D Airplane Datasets

| Encoder | Decoder |
|---|---|
| Input $64{\times}64{\times}3$ image | $3{\times}3$ 1 Conv $\downarrow$ |
| $4{\times}4$ 32 Conv $\downarrow$, BN, LReLU | $4{\times}4$ 1 Conv $\uparrow$, BN, LReLU |
| $4{\times}4$ 32 Conv $\downarrow$, BN, LReLU | $4{\times}4$ 32 Conv $\uparrow$, BN, LReLU |
| $4{\times}4$ 64 Conv $\downarrow$, BN, LReLU | $4{\times}4$ 32 Conv $\uparrow$, BN, LReLU |
| $4{\times}4$ 64 Conv $\downarrow$, BN, LReLU | $4{\times}4$ 64 Conv $\uparrow$, BN, LReLU |
| FC 256 | FC 1024, LReLU |
| FC 2 | FC 256, LReLU |

Table 18: Architecture for the 3D Teapots Dataset

| Encoder | Decoder |
|---|---|
| Input $64{\times}64{\times}3$ image | $3{\times}3$ 1 Conv $\downarrow$ |
| $4{\times}4$ 32 Conv $\downarrow$, BN, ReLU | $4{\times}4$ 3 Conv $\uparrow$, BN, ReLU |
| $4{\times}4$ 32 Conv $\downarrow$, BN, ReLU | $4{\times}4$ 32 Conv $\uparrow$, BN, ReLU |
| $4{\times}4$ 64 Conv $\downarrow$, BN, ReLU | $4{\times}4$ 32 Conv $\uparrow$, BN, ReLU |
| $4{\times}4$ 64 Conv $\downarrow$, BN, ReLU | $4{\times}4$ 64 Conv $\uparrow$, BN, ReLU |
| FC 128 | FC 1024, LReLU |
| FC 6 | FC 128, LReLU |

Table 19: Architecture for the 3D Shape Dataset

| Encoder | Decoder |
|---|---|
| Input $64{\times}64{\times}3$ image | $3{\times}3$ 1 Conv $\downarrow$, Sigmoid |
| $4{\times}4$ 32 Conv $\downarrow$, BN, LReLU | $4{\times}4$ 3 Conv $\uparrow$, BN, LReLU |
| $4{\times}4$ 32 Conv $\downarrow$, BN, LReLU | $4{\times}4$ 32 Conv $\uparrow$, BN, LReLU |
| $4{\times}4$ 64 Conv $\downarrow$, BN, LReLU | $4{\times}4$ 32 Conv $\uparrow$, BN, LReLU |
| $4{\times}4$ 64 Conv $\downarrow$, BN, LReLU | $4{\times}4$ 64 Conv $\uparrow$, BN, LReLU |
| FC 256 | FC 1024, LReLU |
| FC 6 | FC 256, LReLU |

Table 20: Architecture for the 3D Face Model Dataset

| Encoder | Decoder |
|---|---|
| Input $64{\times}64{\times}1$ image | $3{\times}3$ 1 Conv $\downarrow$, Sigmoid |
| $4{\times}4$ 32 Conv $\downarrow$, BN, LReLU | $4{\times}4$ 1 Conv $\uparrow$, BN, LReLU |
| $4{\times}4$ 32 Conv $\downarrow$, BN, LReLU | $4{\times}4$ 32 Conv $\uparrow$, BN, LReLU |
| $4{\times}4$ 64 Conv $\downarrow$, BN, LReLU | $4{\times}4$ 32 Conv $\uparrow$, BN, LReLU |
| $4{\times}4$ 64 Conv $\downarrow$, BN, LReLU | $4{\times}4$ 64 Conv $\uparrow$, BN, LReLU |
| FC 128 | FC 1024, LReLU |
| FC 4 | FC 128, LReLU |

## A.9 System and Model Configurations

All of our experiments were run on a single hardware consisting two DGX2 nodes, collectively consisting of 32-V100 GPUs, 1.5GB GPU RAM, and 3TB System RAM. Encoder and decoder architecture are the same in all experiments. Encoder has two convolutional layers followed by Batch Normalization layer and LeakyReLU activation. After convolutional layers, there is one fully-connected layer with 64 nodes and another layer which maps to the latent space. The decode part is symmetric to the encoder part. $C$ for CCI-VAE is set as 25 for 2D Toy dataset and as 50 for all other datasets.

## A.10 Dataset

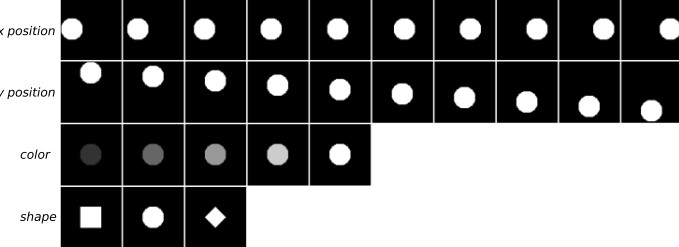

Figure 4: Four factors in datasets. $x$ and $y$ positions have 53 elements, color has 5 elements and shape has 3 elements.

1. 2D Toy Dataset: This dataset has objects with three shapes ($S$) (a circles, a rectangles and a diamonds), and variations to their $x$ and $y$ positions and color information (more specifically, the brightness). This is a rather small, but very effective, dataset. There are 53 unique $x$ positions ($X$), 53 unique $y$ positions ($Y$) and 5 colors ($C$). We create $XY$, $XYC$, $XYS$ and $XYCS$ sub-datasets to show the differences of the latent space when the combination of categorical and continuous factors are presented.

2. 2D Arrow Dataset (Tonnaer et al., 2022): This dataset has $4096$, three-channel RGB, $64 \times 64 \times 3$ images of a 2D object (arrow) with ground truth factors of 64 colors and 64 fixed in-plane rotations.

3. 3D Airplane Dataset (Tonnaer et al., 2022): This dataset also has $4096$, three-channel RGB, $64 \times 64 \times 3$ images of a 3D object (airplane within the ModelNet40 dataset (Wu et al., 2015)) with ground truth factors of 64 colors and 64 rotations with respect to a vertical axis.

4. 3D Teapots Dataset (Eastwood & Williams, 2018): This dataset has $200,000$, three-channel RGB, $64 \times 64 \times 3$ images of a 3D object (teapot) with ground truth factors of independently sampled from its respective distribution: azimuth $\sim U[0, 2\pi]$, elevation $\sim U[0, \pi/2]$, and three colors, namely, red (R), green (G) and blue (B), sampled with $R \sim U[0, 1]$, $G \sim U[0, 1]$, and $B \sim U[0, 1]$. This dataset is very effective to evaluate model when all factors are independently from the uniform distributions.

5. 3D Shape Dataset (Burgess & Kim, 2018): This dataset has $480,000$, three-channel RGB, $64 \times 64 \times 3$ images of 3D objects with ground truth factors of four shapes, eight scales, 15 orientations, 10 floor color, 10 wall colors, and 10 object colors.

6. 3D Face Model Dataset (Paysan et al., 2009): This dataset has $127,050$, gray-scale, $64 \times 64$ images of 3D faces with ground truth factors of 50 different face ids, 21 azimuth, 11 elevation and 11 lighting conditions.

7. Blood Cell Dataset (Acevedo et al., 2020): This dataset has $17,092$, $360 \times 364 \times 3$ images of eight types of normal peripheral blood cells. We used the $28 \times 28 \times 3$ resized dataset followed by Yang et al. (2021).

8. Sprites Dataset (Reed et al., 2015): This dataset has $93,312$, $64 \times 64 \times 3$ images from 1296 unique characters with ground truth factors of the color of hair, tops, body, bottom and actions.

### A.11 LATENT SPACES AND RECONSTRUCTIONS OF LATENT TRAVERSALS ACROSS EACH LATENT DIMENSION

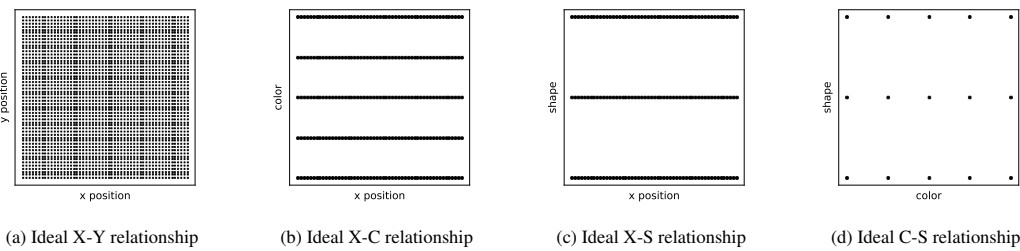

(a) Ideal X-Y relationship  (b) Ideal X-C relationship  (c) Ideal X-S relationship  (d) Ideal C-S relationship

Figure 5: Ideal relationships between X-Y, X-C, X-S and C-S features.

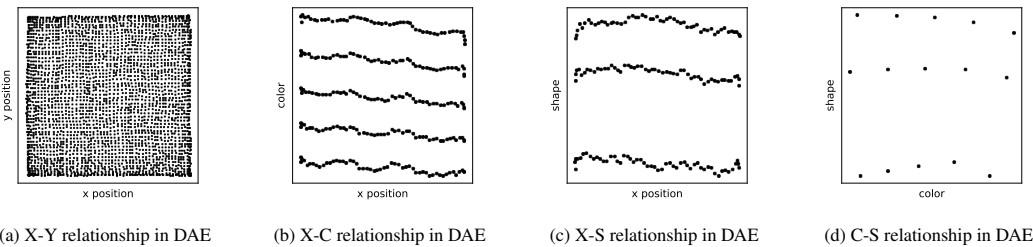

(a) X-Y relationship in DAE  (b) X-C relationship in DAE  (c) X-S relationship in DAE  (d) C-S relationship in DAE

Figure 6: Relationships between X-Y, X-C, X-S and C-S features in DAE.

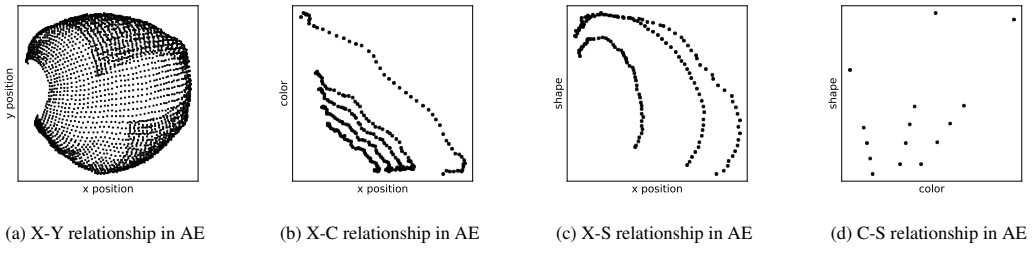

(a) X-Y relationship in AE  (b) X-C relationship in AE  (c) X-S relationship in AE  (d) C-S relationship in AE

Figure 7: Relationships between X-Y, X-C, X-S and C-S features in AE.

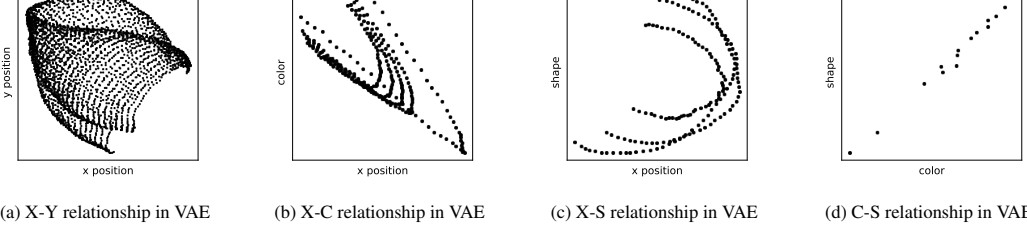

(a) X-Y relationship in VAE  (b) X-C relationship in VAE  (c) X-S relationship in VAE  (d) C-S relationship in VAE

Figure 8: Relationships between X-Y, X-C, X-S and C-S features in VAE.

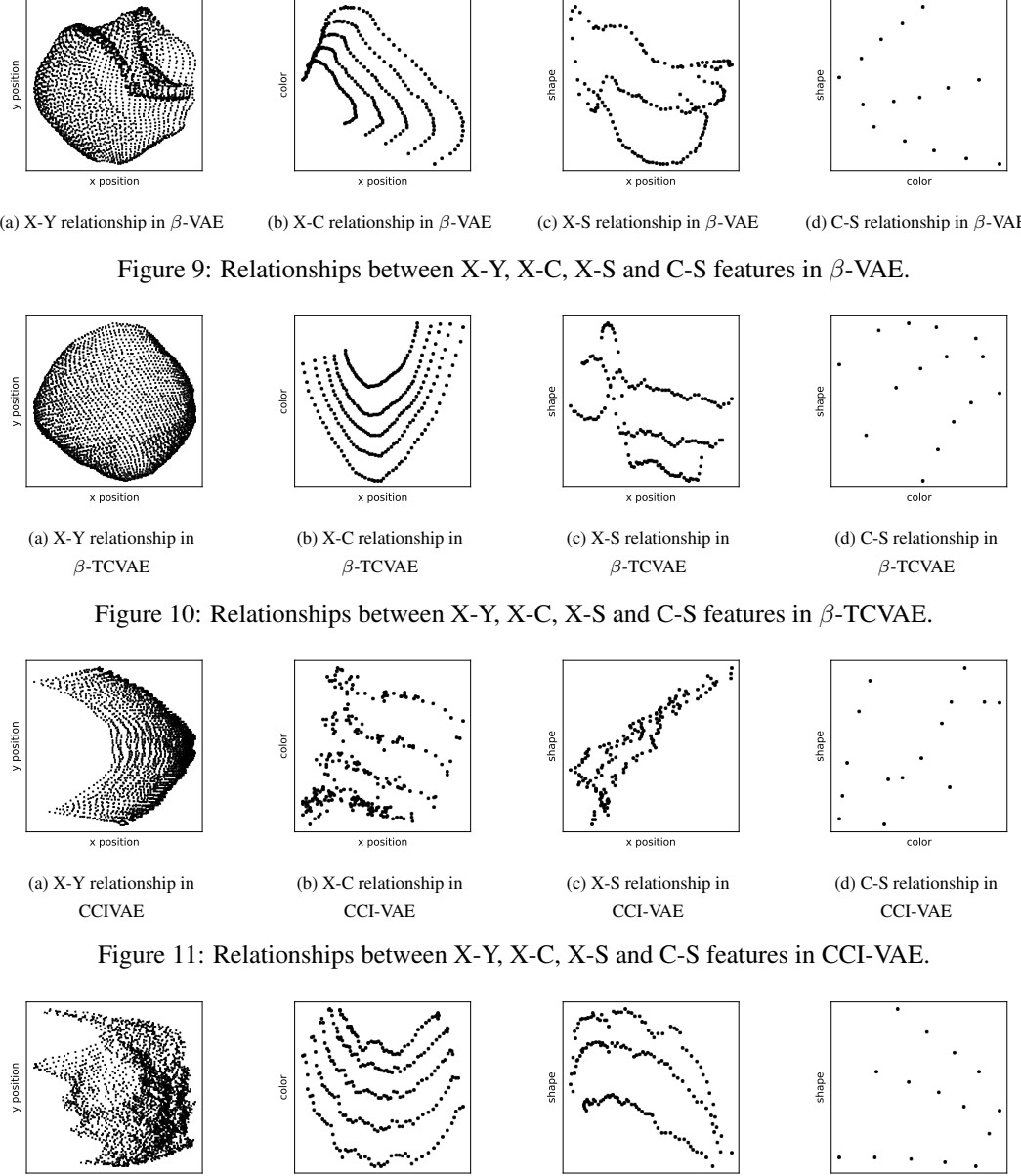

(a) X-Y relationship in $\beta$-VAE    (b) X-C relationship in $\beta$-VAE    (c) X-S relationship in $\beta$-VAE    (d) C-S relationship in $\beta$-VAE

Figure 9: Relationships between X-Y, X-C, X-S and C-S features in $\beta$-VAE.

(a) X-Y relationship in $\beta$-TCVAE    (b) X-C relationship in $\beta$-TCVAE    (c) X-S relationship in $\beta$-TCVAE    (d) C-S relationship in $\beta$-TCVAE

Figure 10: Relationships between X-Y, X-C, X-S and C-S features in $\beta$-TCVAE.

(a) X-Y relationship in CCIVAE    (b) X-C relationship in CCI-VAE    (c) X-S relationship in CCI-VAE    (d) C-S relationship in CCI-VAE

Figure 11: Relationships between X-Y, X-C, X-S and C-S features in CCI-VAE.

(a) X-Y relationship in FVAE    (b) X-C relationship in FVAE    (c) X-S relationship in FVAE    (d) C-S relationship in FVAE

Figure 12: Relationships between X-Y, X-C, X-S and C-S features in FVAE.

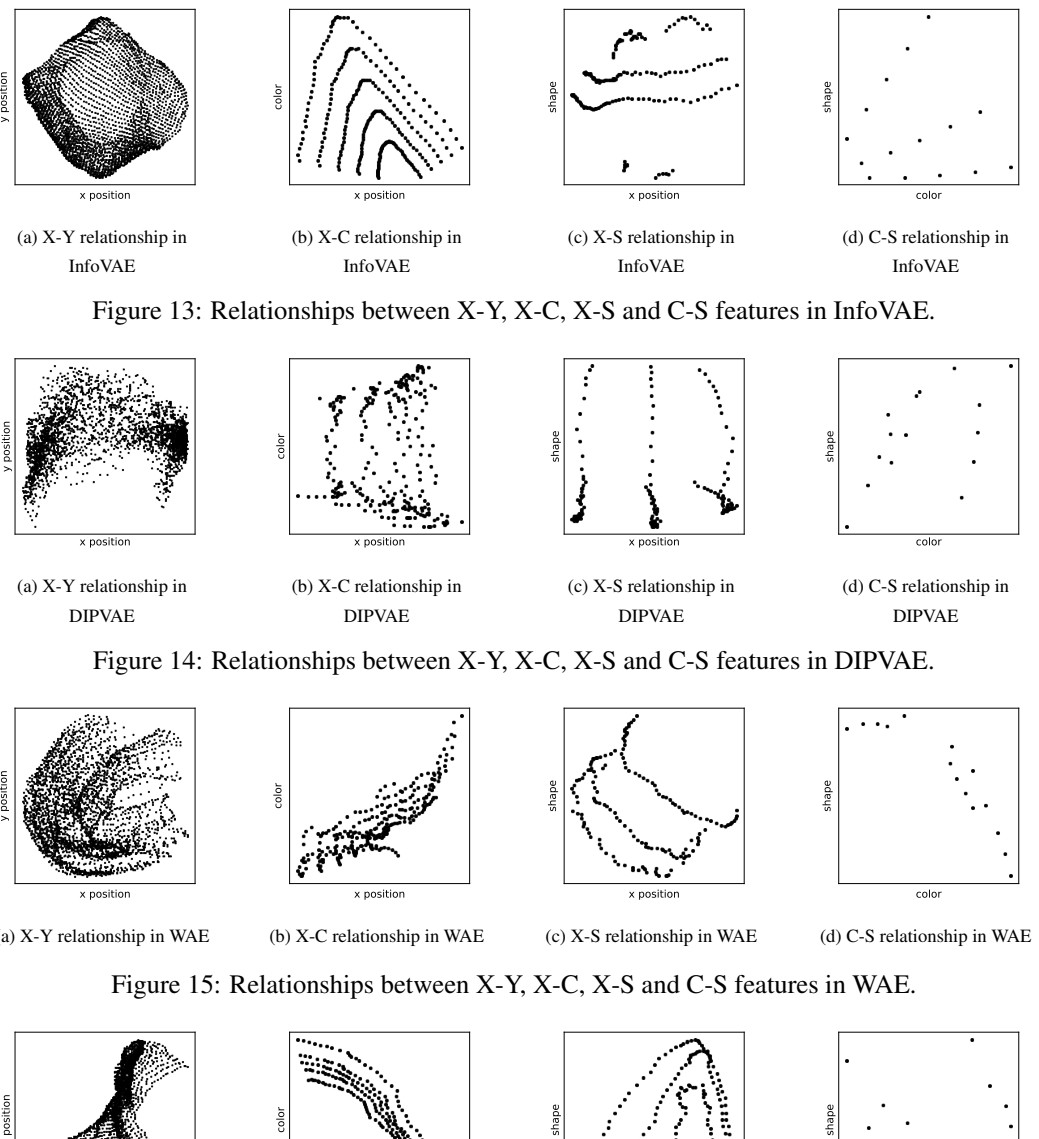

(a) X-Y relationship in
InfoVAE

(b) X-C relationship in
InfoVAE

(c) X-S relationship in
InfoVAE

(d) C-S relationship in
InfoVAE

Figure 13: Relationships between X-Y, X-C, X-S and C-S features in InfoVAE.

(a) X-Y relationship in
DIPVAE

(b) X-C relationship in
DIPVAE

(c) X-S relationship in
DIPVAE

(d) C-S relationship in
DIPVAE

Figure 14: Relationships between X-Y, X-C, X-S and C-S features in DIPVAE.

(a) X-Y relationship in WAE

(b) X-C relationship in WAE

(c) X-S relationship in WAE

(d) C-S relationship in WAE

Figure 15: Relationships between X-Y, X-C, X-S and C-S features in WAE.

(a) X-Y relationship in
LSBDVAE

(b) X-C relationship in
LSBDVAE

(c) X-S relationship in
LSBDVAE

(d) C-S relationship in
LSBDVAE

Figure 16: Relationships between X-Y, X-C, X-S and C-S features in LSBDVAE.

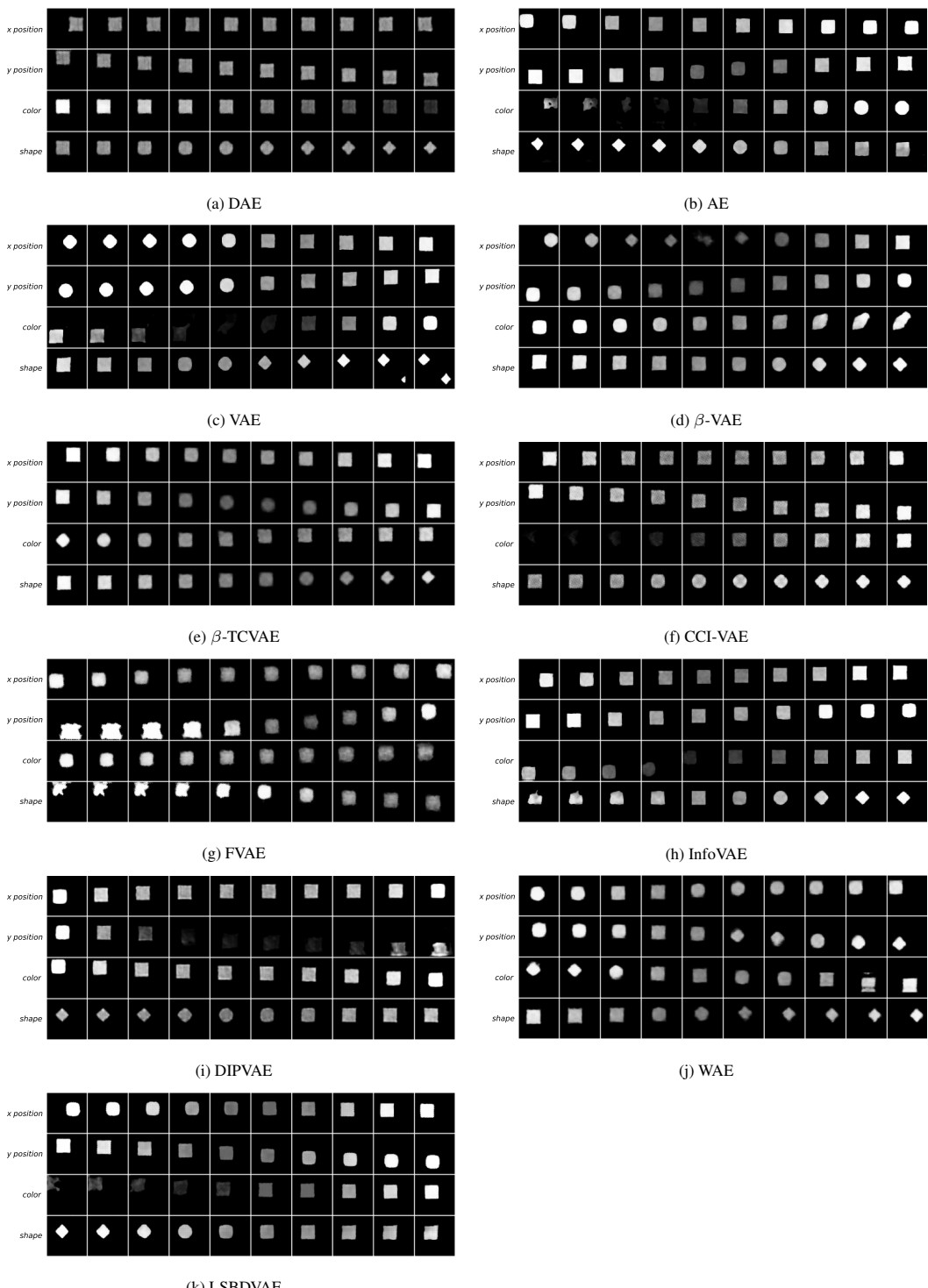

Figure 17: Reconstructions of latent traversals across each latent dimension in the $XYCS$ dataset.

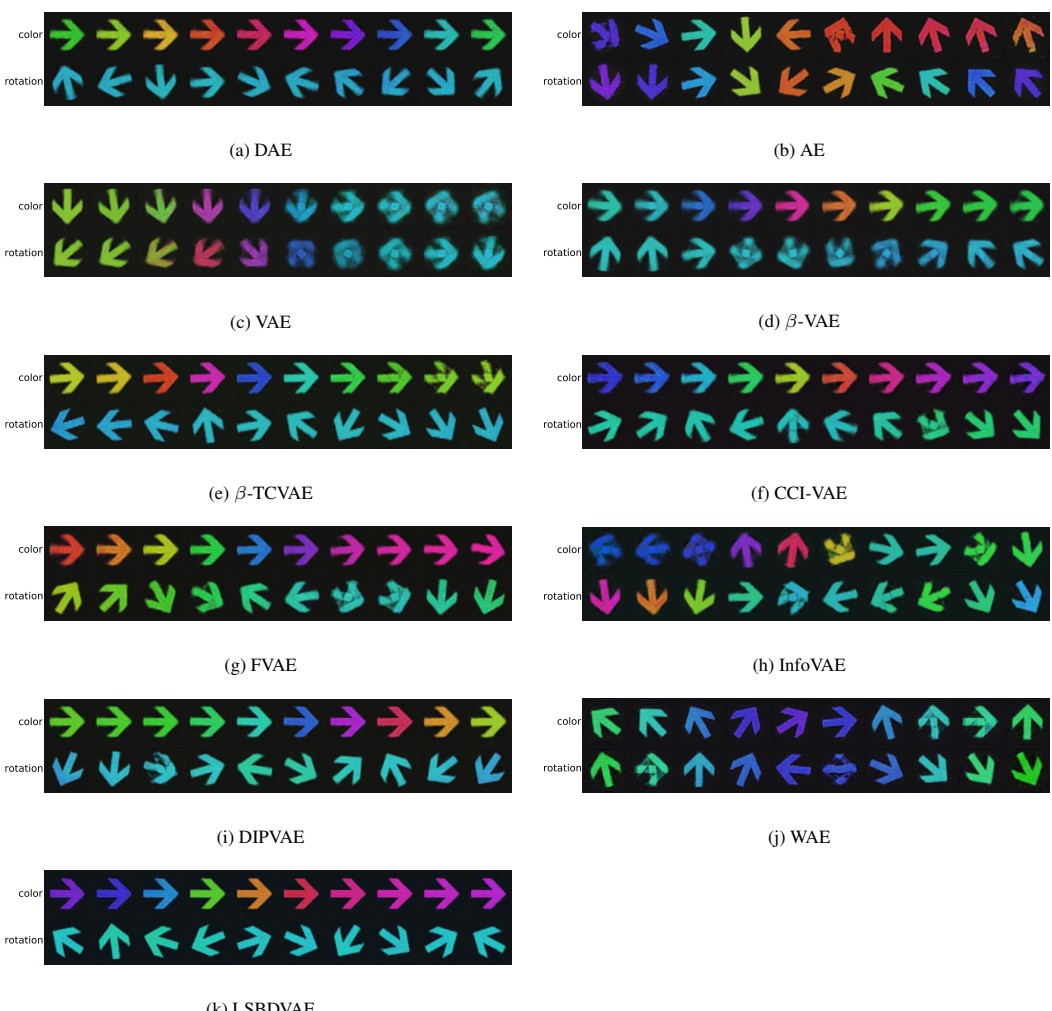

Figure 18: Reconstructions of latent traversals across each latent dimension in the 2D Arrow dataset.

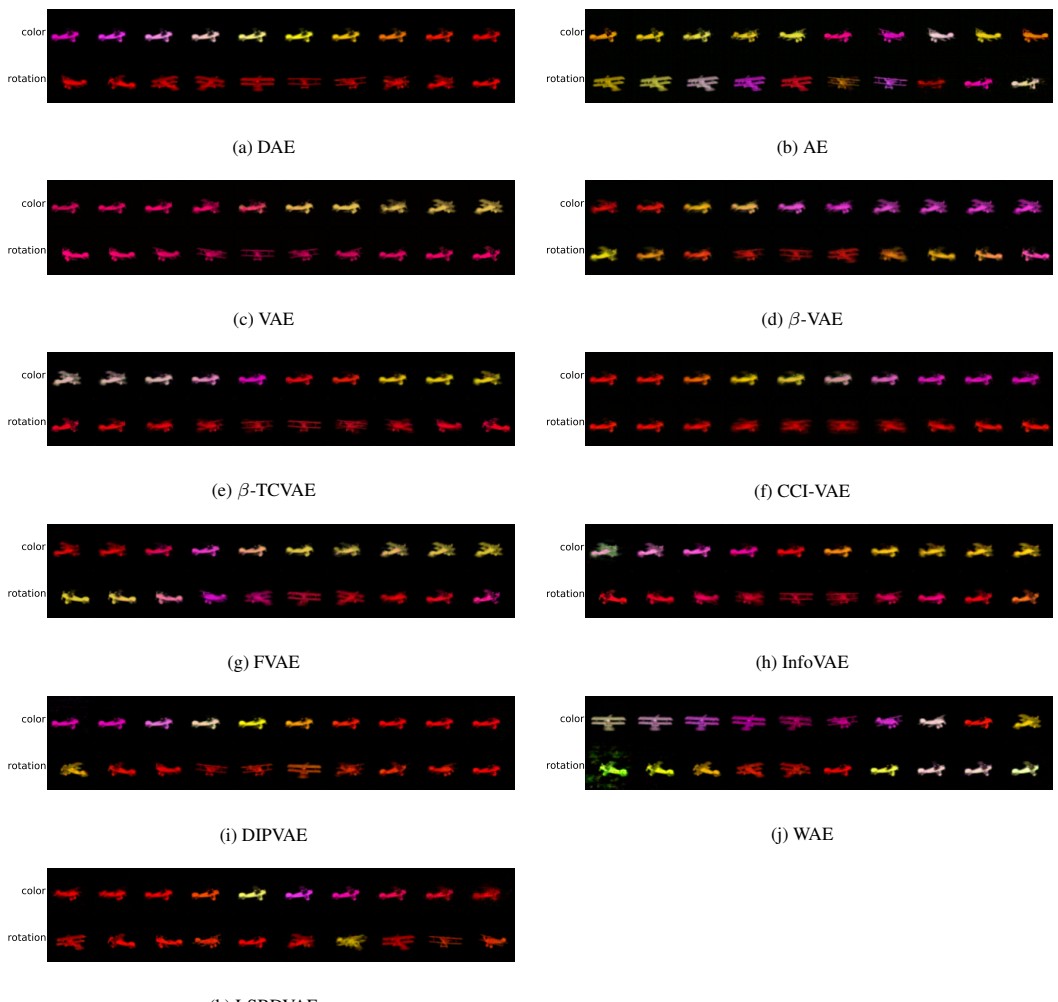

Figure 19: Reconstructions of latent traversals across each latent dimension in the 3D Airplane dataset.

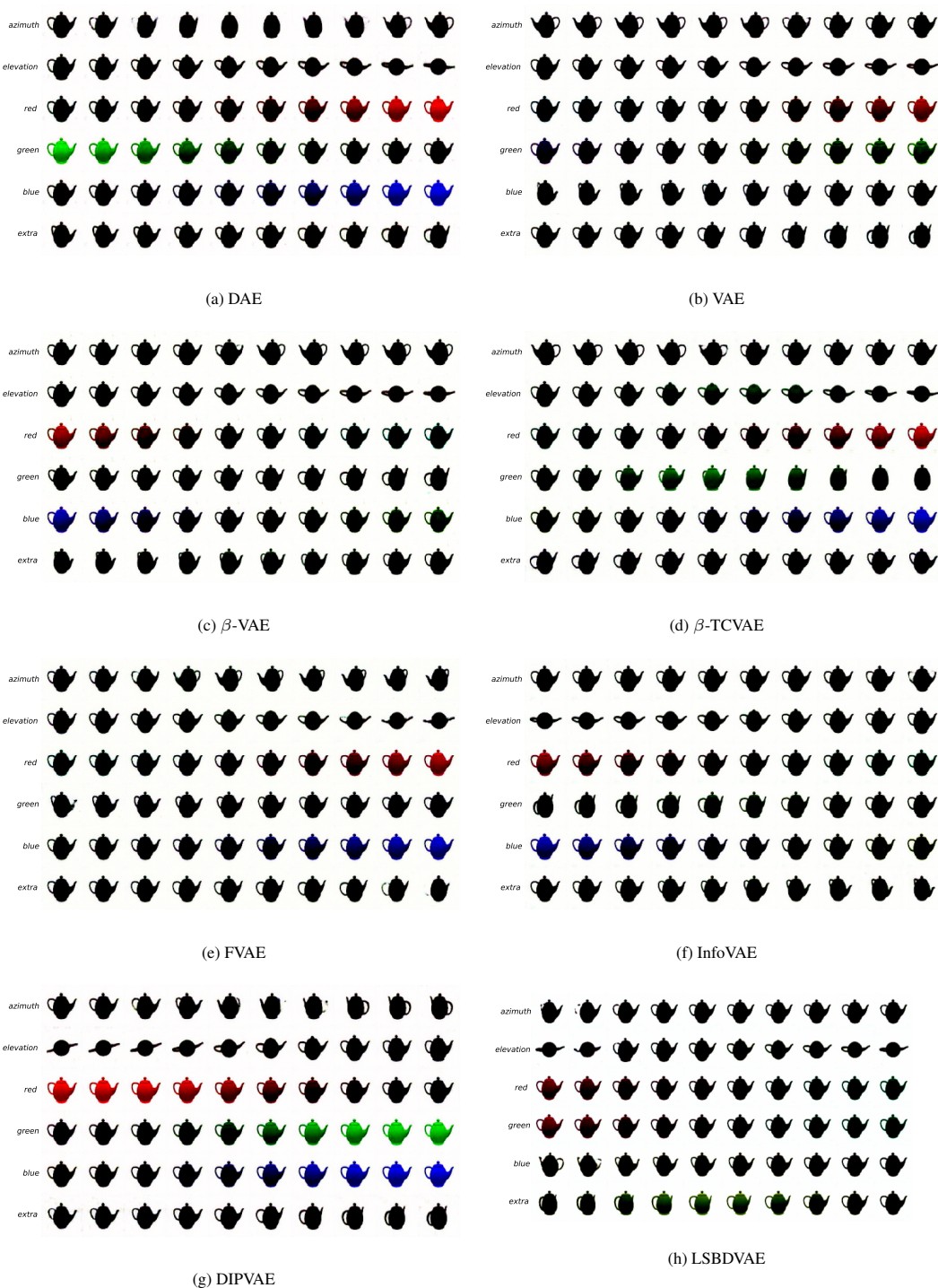

Figure 20: Reconstructions of latent traversals across each latent dimension in the 3D Teapots dataset. Due to the space limit, we omit the result from AE, CCI-VAE and WAE, which have low scores in 3D Teapots dataset.

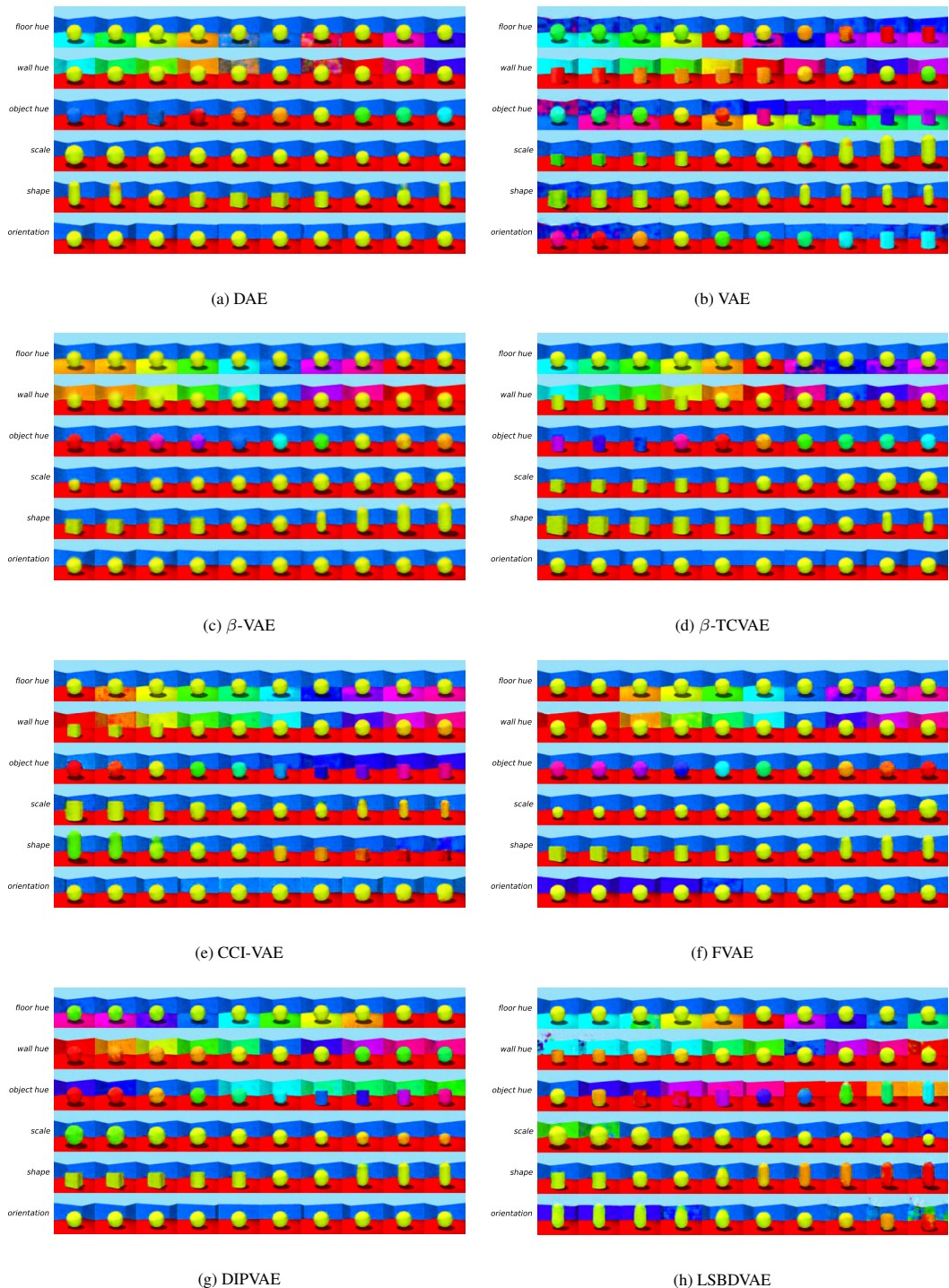

Figure 21: Reconstructions of latent traversals across each latent dimension in the 3D Shape dataset.Due to the space limit, we omit the result from AE, InfoVAE and WAE, which have low scores in 3D Shape dataset.

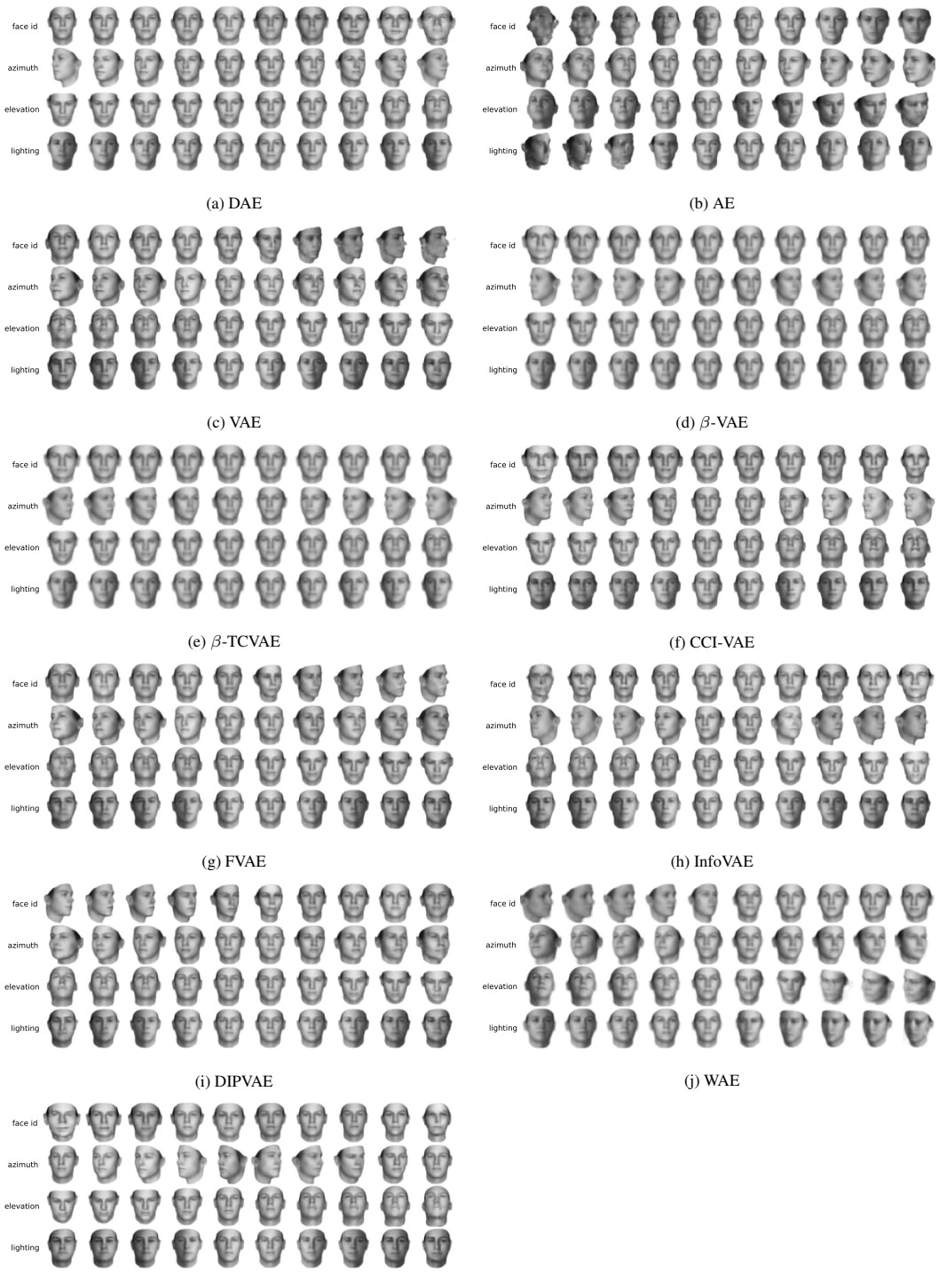

Figure 22: Reconstructions of latent traversals across each latent dimension in the 3D Face Model dataset.

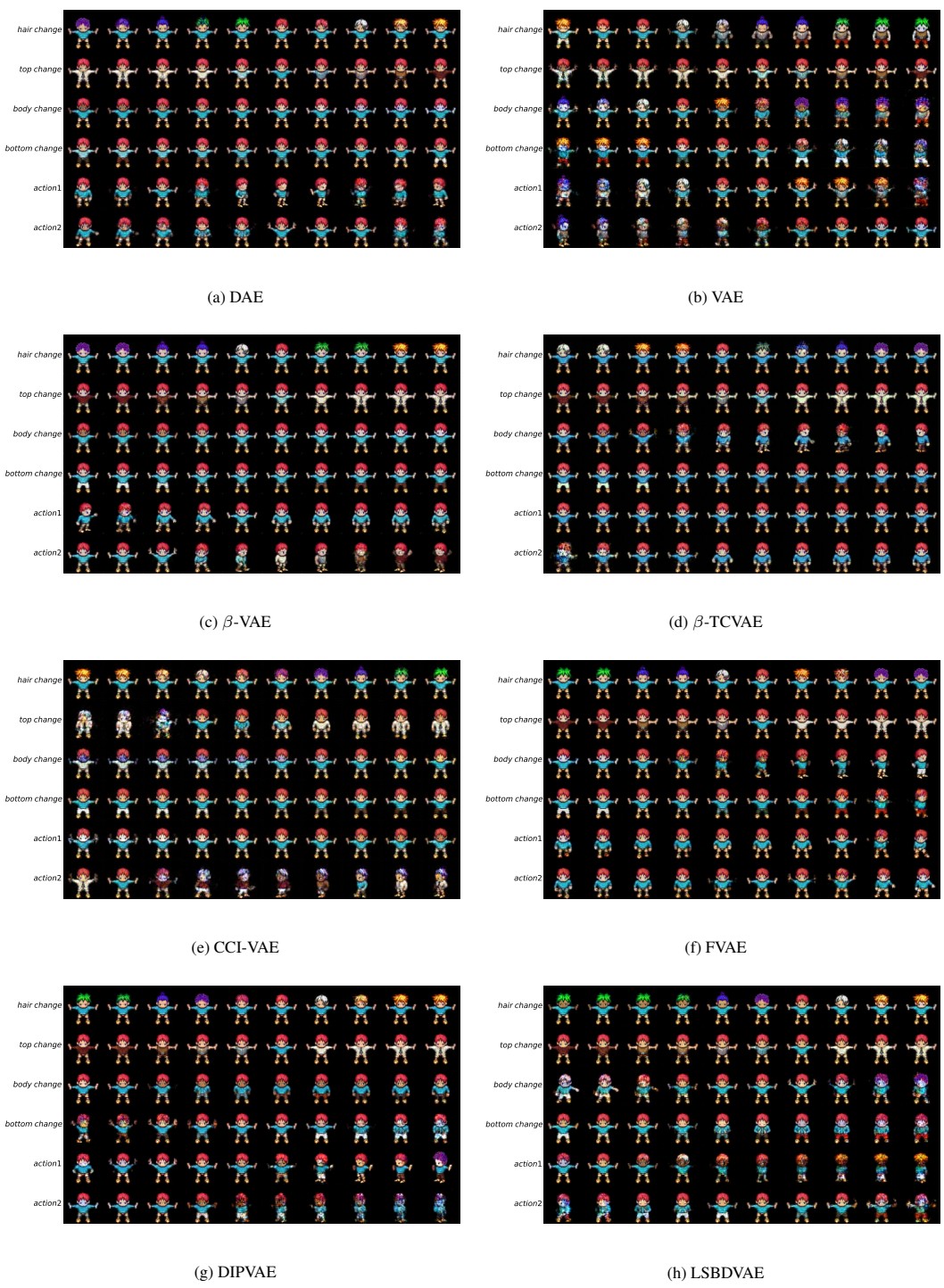

Figure 23: Reconstructions of latent traversals across each latent dimension in the Sprites dataset. Due to the space limit, we omit the result from AE, InfoVAE and WAE. The ground truth factors are the color of hair, tops, body, bottom and actions.

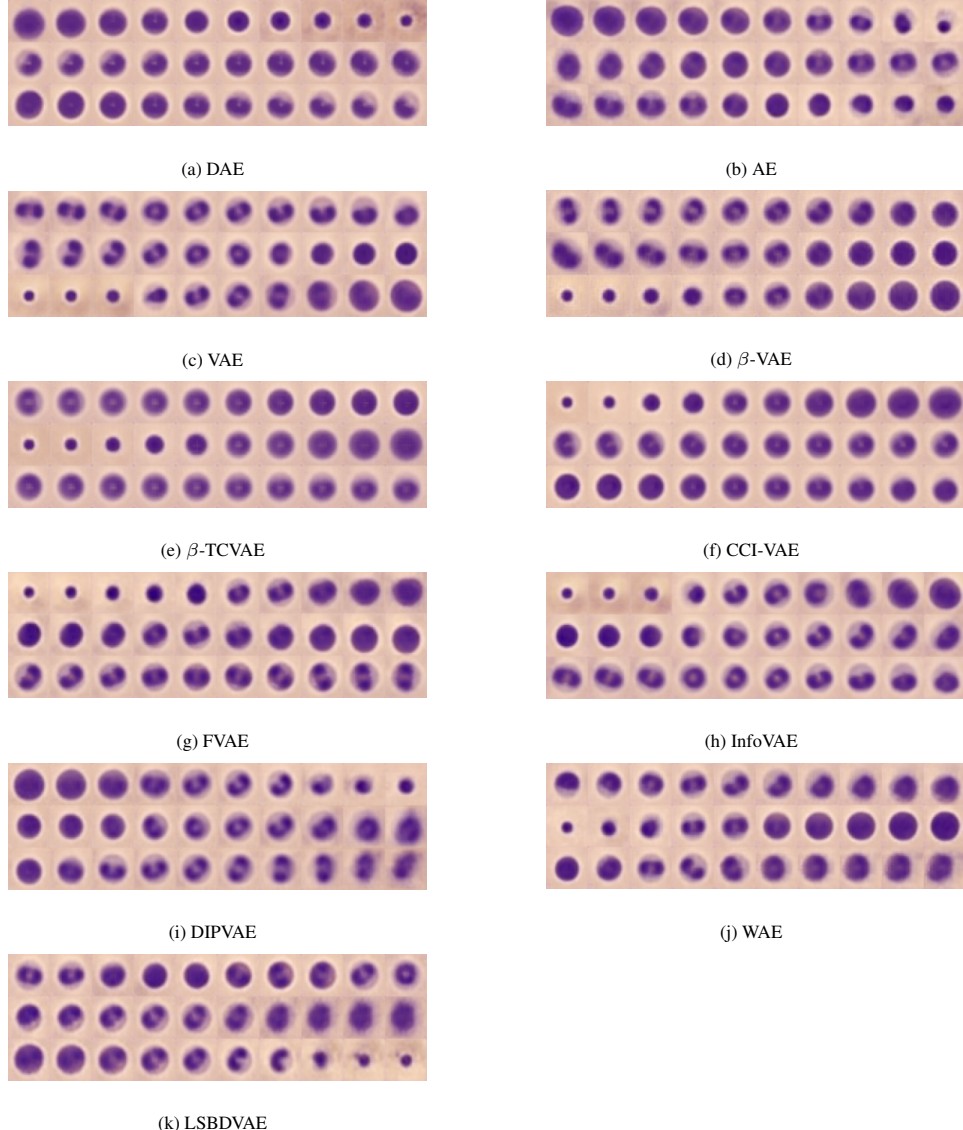

(a) DAE

(b) AE

(c) VAE

(d) $\beta$-VAE

(e) $\beta$-TCVAE

(f) CCI-VAE

(g) FVAE

(h) InfoVAE

(i) DIPVAE

(j) WAE

(k) LSBDVAE

Figure 24: Reconstructions of latent traversals across each latent dimension in the Blood Cell dataset. Some of the models independently learn size and shape of blood cells. For example, for DAE, the first dimension in the latent space corresponds to the size of cells and the second and the third dimensions correspond to changes in shape in different directions.

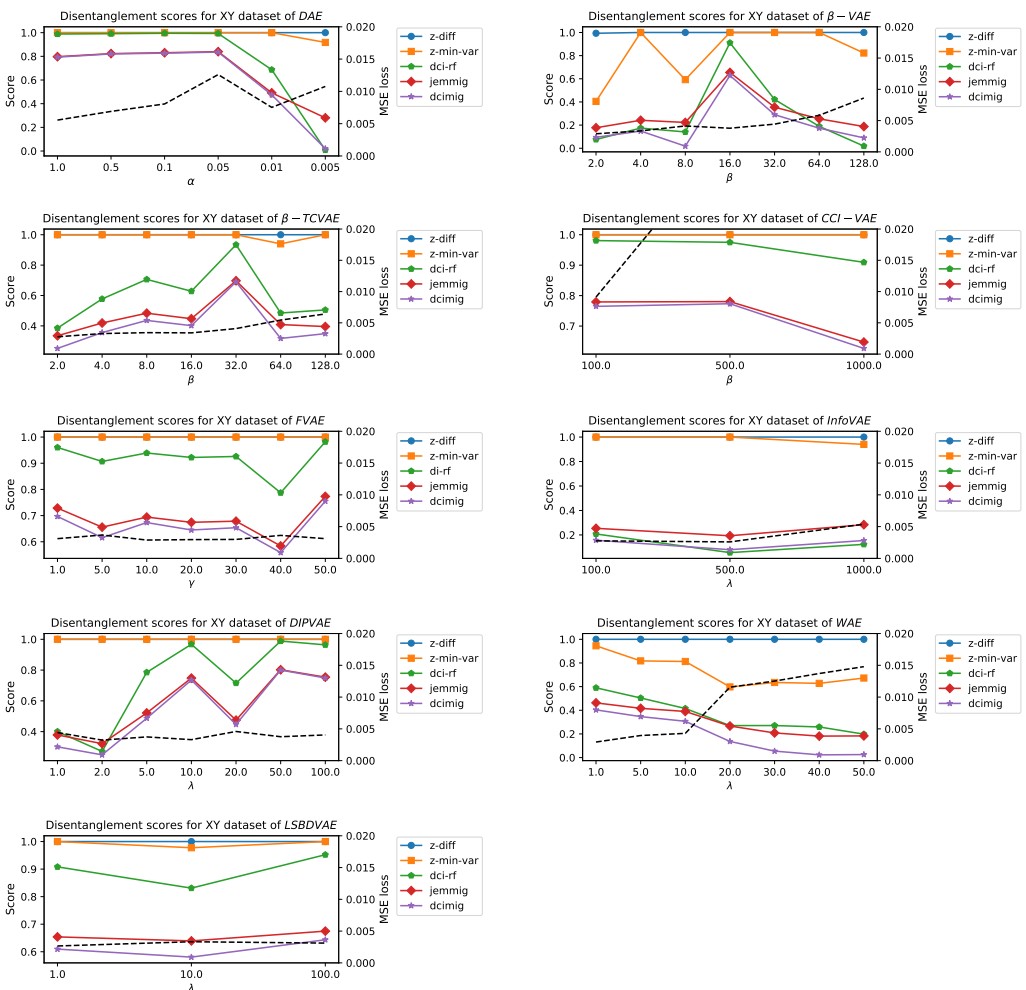

Figure 25: Disentanglement scores with the $XY$ dataset with respect to hyperparameters. For DAE, $w_1$ and $w_2$ obtained by Algorithm 1 are the same in this dataset. Hence, the desirable value for both $w_1$ and $w_2$ is 1. The result shows that when $\alpha$, lower than $0.05$, is assigned to the second dimension, the disentanglement scores also become lower.

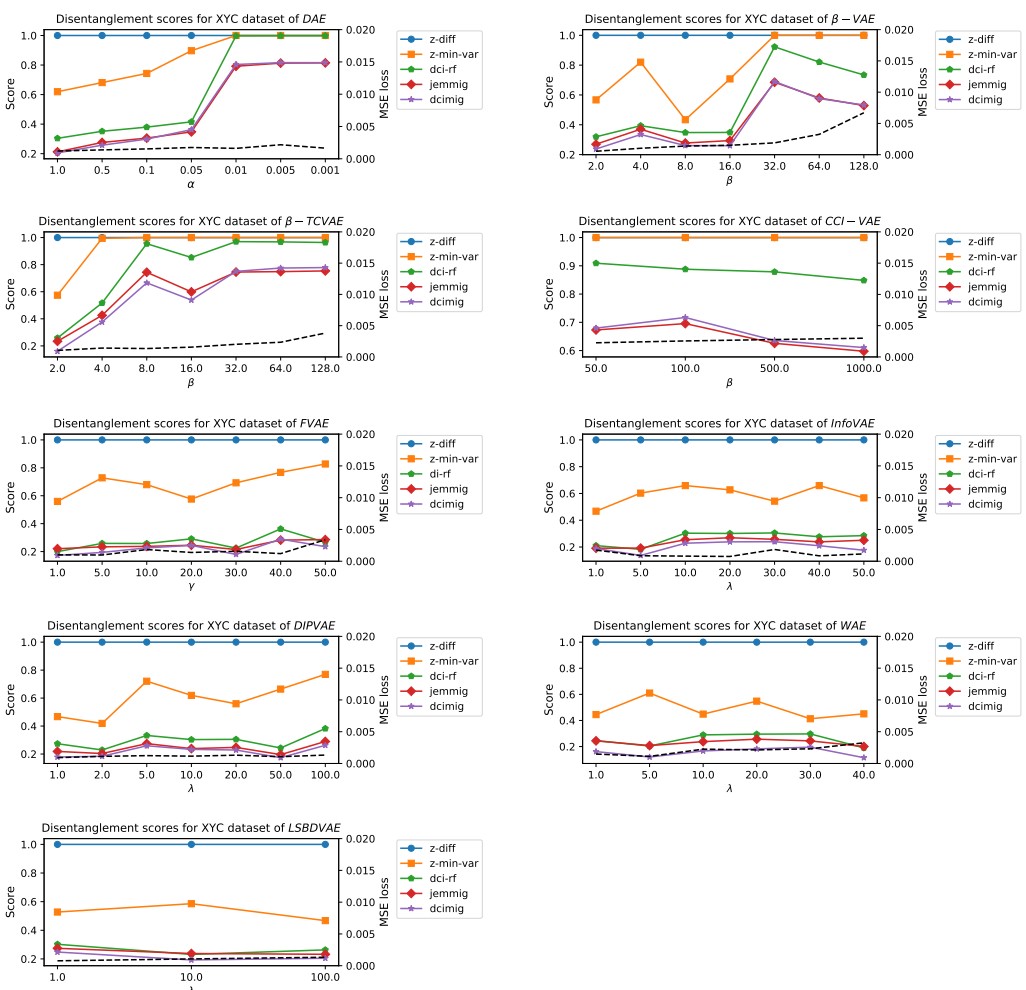

Figure 26: Disentanglement scores with the $XYC$ dataset with respect to hyperparameters.

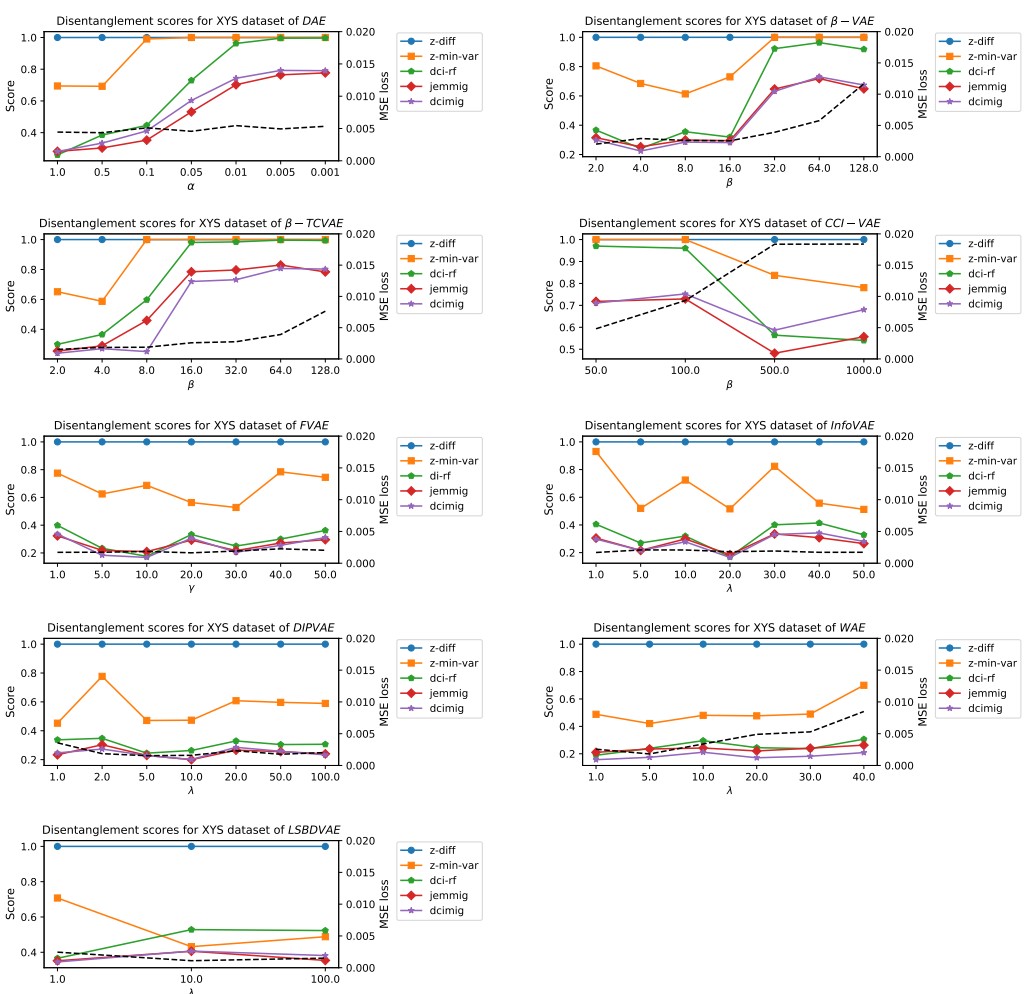

Figure 27: Disentanglement scores with the $XYS$ dataset with respect to hyperparameters.

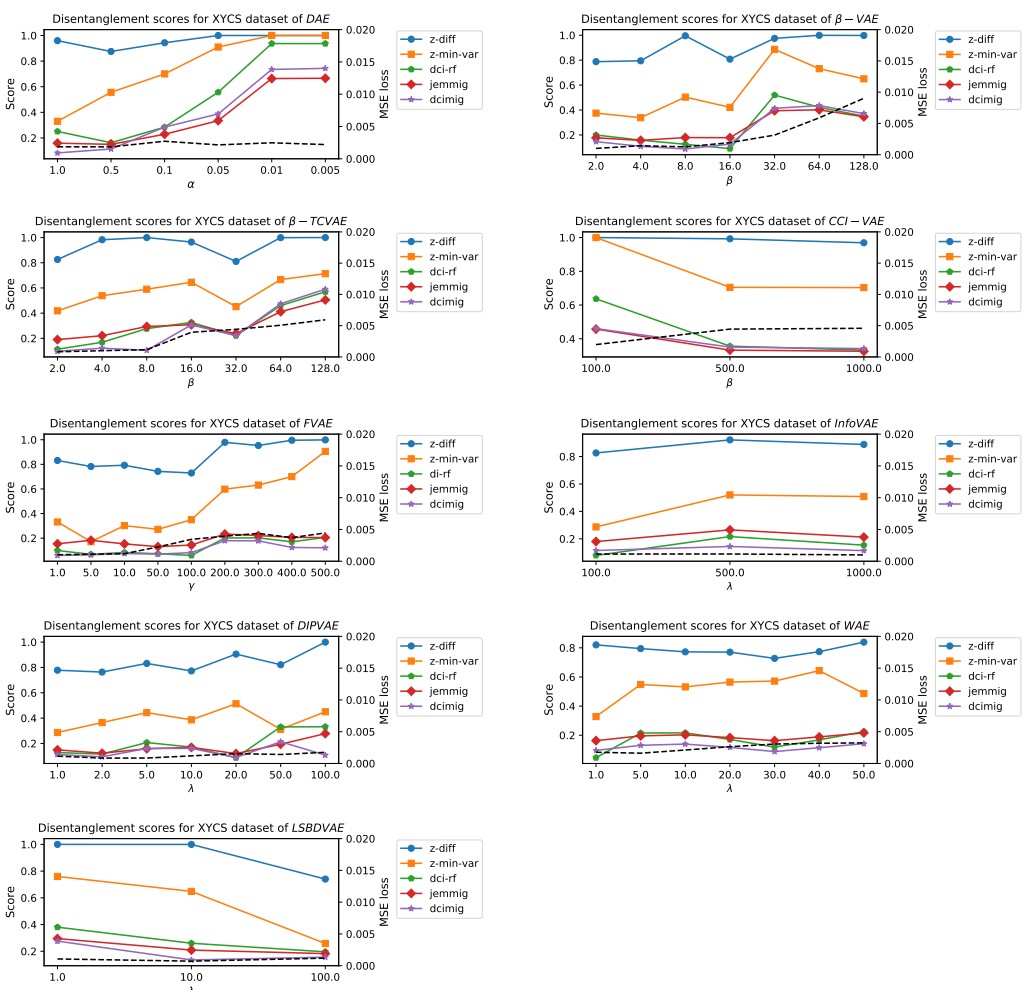

Figure 28: Disentanglement scores with the $XYCS$ dataset with respect to hyperparameters.

