# OpenReview forum: "Learning Disentanglement in Autoencoders through Euler Encoding"
_ICLR.cc/2023/Conference — Submitted to ICLR 2023_

### Official Review · Reviewer_LTGT · 2022-10-19

**Confidence:** 4
**Correctness:** 1
**Technical Novelty And Significance:** 2
**Empirical Novelty And Significance:** 2
**Recommendation:** 3

**Clarity, Quality, Novelty And Reproducibility:**

## Clarity and Quality

### Euler encoding
For starters, the name "Euler encoding" is confusing. Where does "Euler" come from? The closest thing I can think of is Euler's formula in complex analysis. However, the complex exponential function does not appear here. This term was not used in previous work (~164 results from Google), either. If the author coined this term, they should explain why it relates to Euler.

Leaving aside the name issue, it is unclear why the author chose this trigonometric function form. The author stated that the purpose of introducing Euler encoding was "a linear disentangled representation is achieved if changes of components are independent of one another" (inaccurate statement, by the way). However, it is unclear which properties of cos/sin functions can lead to this. The group is not always cyclic, and the latent variable should not necessarily be bounded.

### Disentanglement

The author cited [Locatello et al. 2019] to support the statement, "it is fundamentally impossible to learn disentangled representations without having inductive biases". However, it is unclear how their results can be applied here because their work used a different definition of disentanglement, but this work only focuses on the algebraic aspect.

### Group and group action

It is nice that the author used the algebraic definition of disentanglement. However, I do not think the author ever precisely stated the "clear underlying group structure" anywhere (including the appendix), e.g., which group and group actions were used for each attribute. For example, does the author use a cyclic group for the X/Y position, or do they have some special treatment? Is the group for the shape a cyclic group or an arbitrary permutation group? For 3D objects, was $O(3)$ or $SO(3)$ used? It is hard to evaluate the experimental settings and results without such information.

### Theorems

It is nice that the author tried to analyze the methodology theoretically. However, the theorems are verbose, and I did not see how they can guarantee the proposed model can successfully disentangle the data in an unsupervised fashion.

Theorem 3.1 basically says the composition of equivariant maps is equivariant. This is well-known, but there is a hidden condition. A map is equivariant to two specific group actions, and a group can act on a set in many ways. Say, $b: W \to O$ is equivariant to actions $\mathrm{act}_1$ on $W$ and $\mathrm{act}_2$ on $O$, and $h: O \to Z$ is equivariant to actions $\mathrm{act}_3$ on $O$ and $\mathrm{act}_4$ on $Z$. $f : W \to Z := h \circ b$ is equivariant to $\mathrm{act}_1$ and $\mathrm{act}_4$ if $\mathrm{act}_2$ and $\mathrm{act}_3$ on $O$ are the same.

Then, the paragraphs in p.4 below Theorem 3.1 basically say, "$\mathrm{id}_G$ + an isomorphism $\leadsto$ isomorphic group actions". I do not see how it supports the statement "equivariant map can be learned by an autoencoder" (no matter how the group and group actions are chosen).

Lastly, if I didn't misunderstand anything, we can prove Theorem 3.2 by saying, since $T_i^\alpha: \\{1, \dots, n\\} \times (0, 1) \to Z$ as a function of $i$ and $\alpha$ ($Z \subset \mathbb{R}^n$ is the hypercube), $E: Z \to \mathbb{R}^{2n}$, and $A: \mathbb{R}^m$ are injective, $A \circ E \circ T_i^\alpha$ must be injective. Again, I do not see how it is related to disentanglement.

### Grid

I think the statement "ideal learned latent space should cover a two-dimensional grid" in Figure 1 caption and the assumption "performing independent disentangled actions on a symmetry group causes the corresponding subspace to form a grid-shape latent space" are unjustifiable. Not all group action/representation leads to this. For example, how do you get a group representation of $C_3$ or $D_4$ on a grid (let alone linear)?

### Other issues

- "*To the best of our knowledge, this is the first* deterministic model that <a long description of the model>" sounds redundant. It's called plagiarism if the author already knows they are not the first.
- The author should follow "Formatting Instructions for ICLR 2023 Conference Submissions" and learn to use $\verb|\citet{}|$ and $\verb|\citep{}|$ (Section 4.1).
- "The quality of the disentanglement is fully reliant on ideal or near-ideal priors" has no support.
- It's unclear why "values smaller than unity are replaced with hyperparameter $\alpha$" and what "unity" means.
- Does "Gaussian interpolation is used to map unseen examples to known examples" mean that the model has completely no generalization ability?
- The meaning of the interpolation layer (Algorithm 2) and "weight-sensitive Gaussian noise" is unclear.

## Novelty

To the best of my knowledge, the so-called "Euler encoding" (Eq. 6) is novel. Other aspects of the model exist in prior work.

## Reproducibility

The author provided a brief review of the encoder/decoder architectures but did not provide any training details, such as hyperparameters. No code was found.

**Strength And Weaknesses:**

## Strengths

- Unsupervised disentanglement is an important problem.
- The author used the algebraic definition of disentanglement.
- The author compared the proposed architecture with many autoencoder-based methods.

## Weaknesses

- The author tried to support the proposed method with theoretical guarantees, but the theorems seem not to guarantee the learned representation is disentangled.
- The proposed metric, Grid Fitting Score, relies on the assumption that "performing independent disentangled actions on a symmetry group causes the corresponding subspace to form a grid-shape latent space", which may not hold (details below).
- The groups and group actions in experiments are not given, so the experimental settings are a bit questionable.


**Summary Of The Paper:**

This paper proposed an **autoencoder** architecture based on **trigonometric functions** for **unsupervised disentangled representation learning**, introduced a new disentanglement metric (based on a strong assumption), and evaluated the proposed architecture on six datasets.

**Summary Of The Review:**

The author tried to solve a challenging problem (unsupervised disentanglement) and conducted abundant experiments. However, the grid assumption is unreliable, their theory does not support the proposed method, and the detailed group theoretical treatment in experiments is unclear. Therefore, I cannot recommend this paper for acceptance to ICLR.

---

> ### Author Response · Authors · 2022-11-13
> **Response to Reviewer LTGT -1**
>
> Thank you for the feedbacks and your questions. To facilitate the response to your questions, first, we would like to make the problem statement even more clear.
>
> Let $G$ is generated by a set $S=\\{s_1,s_2,⋯,s_n\\}$ subject to a set of $R$ of relations among elements in $S$. Suppose we have a group action $\cdot ∶G\times W→W$. Then we say that the action is disentangled by the relations $R$ if there is a decomposition $W=W_1\times W_2 \times \cdots \times W_n$ and actions $\cdot_i:〈s_i 〉 \times W_i \rightarrow W_i, i \in \\{1,…,n\\}$ such that:
>
> (1) $(s_1^{\epsilon_1},…,s_n^{\epsilon_n} )\cdot(w_1,…,w_n )=(s_1^{\epsilon_1}\cdot w_1,…,s_n^{\epsilon_n}\cdot w_n )$ for all ${\epsilon_i}\in Z$ and $w_i\in W_i$ and
>
> (2) if any element $g\in G$ can be written uniquely in the form $g=s_1^{\epsilon_1} \cdots s_n^{\epsilon_n}$ for some ${\epsilon_i}\in Z$ by the relations $R$.
>
> The aim of this paper is to learning the actions $\cdot_i:〈s_i 〉×W_i\rightarrow W_i$ in the latent space, $Z$, by mapping $(s_1^{\epsilon_1} ,…,s_n^{\epsilon_n})$ to $(e^{i\alpha_i \epsilon_1},…,e^{i\alpha_n \epsilon_n})$ and $W_i$ to $Z_i$ for some $\alpha_i \in (0,1)$. However complex numbers are undesirable in machine learning and so we use a pair of cosine and sine functions, which is Euler encoding.
>
> With this rephrased problem statement, let us respond to your questions.
>
> 1. Euler encoding
>
> As you have summarised, the term Euler encoding refers to Euler’s formula in complex analysis. This is because we are interested in mapping $s_j^{\epsilon_j}\cdot w_j$ to $e^{i(\alpha_j \epsilon_j +z_j)}$, and nth root of unity is a cyclic group. However complex numbers are undesirable in machine learning and so we use a pair of cosine and sine functions which can be seen as Euler’s formula.
> To make the name or the intuition clear, we have modified our paper to clarify where this “Euler encoding” comes from.  Thank you for pointing this out.
>
> 2. Disentanglement
>
> Even though [Locatello et al. 2019] used a different definition of disentanglement, the key intuition of the disentanglement that “a change to a single underlying factor of variation should lead to a change in a single factor in the learned representation”, remains the same.
> Furthermore, the Theorem 1 in [Locatello et al. 2019] argues that the challenge of disentanglement when there are many different ways to express underlying factors. As such, we cannot distinguish the representations from disentangled factors and entangled factors.
> In our problem statement, since we have different ways to express any element $g\in G$ from the generating set $S$ subject to the relations $R$, the assumption still holds true and hence the results are applicable in our context as well. We have now made this clear in the modified version of the manuscript.
>
> 3. Group and group action
>
> From the problem statement above, we aim to learn cyclic groups which is generated by single element in $S$. We hope this clears all other concerns in this question. We have made this clearer in the modified version of the manuscript.
>
> 4. Theorem
>
> Thank you for pointing out the details. Although there are many ways a group can act on a set, we aim to learn disentangled actions by the relations R that we proposed in the problem statement. To learn the disentangled actions, we need to learn both a process of generation and inference. Since an autoencoder architecture can mimic a generation process by a decoder, and an inference process by an encoder, we mentioned that the equivariant map ‘can’ be learned by an autoencoder.
>
> In addition, thank you for suggesting a different way to prove Theorem 3.2. In the proof of the Theorem 3.2, we show that $E(T_i^\alpha (z))=S_i^\alpha \cdot E(z)$. Since $S_i^\alpha$ is orthogonal, it preserves orthogonality, which enables the changes of output from changes of different latent dimensions to be orthogonal.
>
> In summary, the proposed method based on autoencoder, which an unsupervised learning technique, with Euler layer can learn disentanglement in an unsupervised fashion by preserving orthogonality in the latent space through the Euler layer.

---

> ### Author Response · Authors · 2022-11-13
> **Response to Reviewer LTGT-2**
>
> 5. Grid
>
> Thank you for raising this question. Since the goal of learning disentanglement is to make a change in a single underlying factor of variation that would lead to a change in a single factor in the learned representation, if we can achieve disentangled actions, each dimension should be orthogonal. With the refined version of the problem statement, this is easier to see. For example, for $D4$, we want to learn r (rotation) and f (reflection). Since any element in $D4$ can be uniquely written as $r^i f^j$, if we map $r^i$ and $f^j$ to the first and the second dimensions of the latent space, respectively, we have a grid shape latent space.
>
> As such, we politely would like to stay firm on our argument, and indeed it is justifiable to say that "performing independent disentangled actions on a symmetry group causes the corresponding subspace to form a grid-shape latent space".
>
>
> 6. Other issues
>
> (1) Thank you for raising this concern, and we politely would like to highlight that this is why we use the phrase, “to the best of our knowledge”
>
> (2) Thank you for pointing out this. We have changed all of them.
>
> (3) Probabilistic models have a regularizer to minimize the difference between the approximate posterior and the prior. Hence, we mentioned that the quality of the disentanglement of VAE-based approaches are probabilistic is fully reliant on ideal or near-ideal priors. We also agree that the quality of the disentanglement depends on the other aspects. So, we have modified this in our paper.
>
> (4) Thank you for pointing out this. Unity is a singular term meaning “one”. We have clarified this in our paper.
>
> (5) The model is generalizable since we have normalization layers. However, we use Gaussian interpolation layer to further generalize the network since the proposed model is deterministic.
>
> (6) We have updated the Algorithm 2 make this clear.

---

### Official Review · Reviewer_9cKh · 2022-10-24

**Confidence:** 2
**Clarity, Quality, Novelty And Reproducibility:** The paper is clearly written and seem…
**Correctness:** 3
**Technical Novelty And Significance:** 3
**Empirical Novelty And Significance:** 2
**Recommendation:** 6

**Strength And Weaknesses:**

The paper explains the new framework in detail and compares against other autoencoder models on a range of disentanglement metrics.
At the same time, all the comparisons are against other autoencoder models and ignores other types of unsupervised deterministic models e.g.
Learning the Multilinear Structure of Visual Data
M Wang, Y Panagakis, P Snape, S Zafeiriou
CVPR 2017
The current formulation does not cover the case of disentanglement with missing data.
Evaluations are based purely on disentanglement metrics and it is not clear how much difference in downstream tasks some gains achieve.

**Summary Of The Paper:**

The paper proposes a non-probabilistic approach to use autoencoder for learning disentanglement in unsupervised manner. It also proposes a new metric to quantify the disentanglement.

**Summary Of The Review:**

Overall the paper proposes a novel framework to address the problem of disentanglement with unsupervised data. Though the paper is overall sound, the evaluation could be improved by highlighting the differences achieved on some downstream tasks as it's unclear whether the gains are significant.

---

> ### Author Response · Authors · 2022-11-13
> **Response to Reviewer 9cKh**
>
> 1. Other model comparison
>
> Thank you for the references. We agree that there are still many disentangling methods, but, understandably it is a monumental task to compare all of them. Therefore, we focus only on ten models that do not require any labels or pairs of inputs and are entirely autoencoder-based.  We have now made this clear in our modified version of the manuscript.
>
> 2. Downstream tasks
>
> The goal of this paper is to show that the proposed architecture achieves better disentanglement on different datasets introducing inductive bias into the model and data and compared it with different autoencoder-based models to open several opportunities for linear disentangled representation learning based on deterministic autoencoders. However, we agree that it is an interesting direction to see the effectiveness of the model for a number of downstream tasks, and we intend to explore this in our future work. The new dataset we have included in our evaluation, Blood Cell [1] in the Appendix due to the space limit (see Figure 24), shows the benefit of this for downstream tasks like classification.
>
>
> [1] A dataset of microscopic peripheral blood cell images for development of automatic recognition systems," Data in Brief, vol. 30, pp. 105474, 2020.

---

### Official Review · Reviewer_T3ur · 2022-10-25

**Confidence:** 4
**Correctness:** 3
**Technical Novelty And Significance:** 3
**Empirical Novelty And Significance:** 2
**Recommendation:** 5

**Clarity, Quality, Novelty And Reproducibility:**

- The idea that introduces the Eular layer for unsupervised disentanglement learning is interesting and quite novel.
- The paper is clearly written and easy to follow.
- Ablation studies and detailed analyses are missing.
- Although the experimental validation is quite good, the applicability for real-world data is not demonstrated.


**Strength And Weaknesses:**

Strength
- I think designing good explicit inductive biases is particularly important for unsupervised disentanglement learning. The architectural design that enforces the neworks to learn disentangled features can resolve many difficulties in previous methods: For example, the necessity of independence-inducing regularization loss (which can cause the tradeoff between disentanglement and reconstruction) can be removed. In this aspect, I believe this work addressed an important problem and did a good job.
- I feel the Euler encoding is technically sound and the mathematical notations and derivations are quite solid.
- The experiments are comprehensive in terms of the number of datasets and evaluation metrics.
- The paper is well backed-up by supplementary material that contains sufficient details of the method and experiments and presents additional results.

Weakness
- I was not able to find ablation studies and/or in-depth discussions to argue the effectiveness of the three elements described in Section 3.3. For example, how did the authors set the hyperparameter alpha in Algorithm 1 and what happens with varying the value of alpha? What happens if different normalization schemes are used or any normalization is not applied? Is the Gauassian interpolation layer indeed necessary? It would be very helpful to add experimental/theoretical support regarding them.
- The architectures used in the experiments consist of vanilla convolutional and transposed convolutional layers with relatively shallow depths. I feel they are a bit outdated; I would recommend the authors to include experimental results using the networks with residual blocks and deeper depths.
- My major concern is the applicability for real-world datasets; the proposed method was validated melely on simulated datasets. Although many unsupervised disentanglement algorithms also face this issue regarding their practical usage, it would be very helpful to present visual disentanglement results on CelebA or other real-world images to show the potential of the proposed method.

**Summary Of The Paper:**

This paper proposes a deterministic VAE for unsupervised disentanglement learning with Euler encodings. By introducing an architectural inductive bias with the Eular layer, this approach can learn disentanglement without using independence regularization terms or additional supervisions. In addition, the authors present several ways to tackle practical issues with PCA, min-max normalization, and Gaussian interpolation. An interesting disentanglement metric that does not need label information is also proposed. Experiments were conducted on several benchmarks with different metrics.

**Summary Of The Review:**

The main idea is quite good and this paper includes several interesting points. However, the in-depth analyses of the proposed method are missing and thus the sources of good performance are not identifiable. Furthermore, the empirical support is weak in terms of the applicability for real-world data. I thus find it difficult to argue for acceptance of the work.

---

> ### Author Response · Authors · 2022-11-13
> **Response to Reviewer T3ur**
>
> 1. Three elements described in Section 3.3
>
> We varied the hyper-parameter alpha like any other hyper-parameter variants of VAE. With Figures 25-28, we show that the proposed model maintains the reconstruction loss as small as possible whilst offering improved disentanglement scores when we change the alpha. The goal of the normalization layer is to ensure that each feature falls within the (0,1). As normalization satisfies this requirement, the proposed model will work. In cases where this is not the case, the model will first fail to satisfy the theorem 3.2, at which point we know that the disentanglement will fail. This is one of the reasons we use the simplest dataset, namely, XY dataset, to demonstrate and the disentanglement. We compare the results without the normalization layer in the Table below:
>
>
> |Metric	|All together	|Without a normalization layer|
> | ------------- | ------------- | ------------- |
> |z_diff	|1.00	           |1.00|
> |z_var	|1.00	           |1.00|
> |dic_rf	|0.99        	|0.95|
> |jemmig	|0.90	          |0.71|
> |dcimig	|0.90	           |0.70|
>
>  We have now added all results when we remove the Euler layer or the normalization layer in the Appendix (see Table 11). From the results in Table 11, we can clearly see that the disentanglement scores drop significantly without the Euler layer or the normalization layer as dataset has more features.
>
> Finally, the Gaussian interpolation layer, leads to small changes to the overall score for simple datasets, the changes are considerably noticeable for complex datasets. We attribute this to the smoothing effect of the latent space. We have modified our manuscript to state these aspects clearly. We also intend to evaluate the impact of different interpolation techniques in our future work.
>
> 2. Model architecture
>
> The goal of this paper is to show that the proposed architecture achieves better disentanglement on different datasets introducing inductive bias into the model and data and compared it with different autoencoder-based models. For a fairness, we used the most common architectures used in the previous literatures in [1-4].
>
> [1] beta-vae: Learning basic visual concepts with a constrained variational framework, 2017
> [2] Disentangling by factorising, 2018
> [3] Learning disentangled representations and group structure of dynamical environments, 2020
> [4] Quantifying and learning linear symmetry-based disentanglement, 2022.
>
> Although using deeper models is possible, that would potentially make our model perform unreasonably and unfairly better than existing methods, and as such, the comparison here would be unfair.
>
> 3. Real-world images
>
> Thank you for the suggestion. We have now added a new result from blood cell [1] in the Appendix due to the space limit (see Figure 24). Although this dataset does not provide underlying group structures, there are two key factors to classify blood cells 1) size and 2) shape. We visually show that the proposed models learn size and segmentation process of cells independently. In addition, we also have added the Sprites dataset [2] in the Appendix (see Figure 23), which is more complex and other papers submitted to this ICLR 2023 conference  used.
>
> [1] A dataset of microscopic peripheral blood cell images for development of automatic recognition systems," Data in Brief, vol. 30, pp. 105474, 2020.
>
> [2] Deep visual analogy-making. Advances in neural information processing systems, 28, 2015.

---

### Official Review · Reviewer_JNBF · 2022-11-02

**Confidence:** 4
**Correctness:** 3
**Technical Novelty And Significance:** 3
**Empirical Novelty And Significance:** 3
**Recommendation:** 6

**Clarity, Quality, Novelty And Reproducibility:**

The paper is overall well written and clear. The idea of using Gram matrices to do the clustering in the first state is novel and also leads to improved performance. The author also provides a detailed  theoretical analysis.


**Strength And Weaknesses:**


Strengths:
1. The author proposed a new perspective for learning disentangled representation, which introduces the Euler encoding to force the latent space to achieve a linear disentangled representation.
2. The author provides a detailed theoretical analysis.
3. The proposed method achieves good results on toy datasets.


Weaknesses:
1. Some classic metrics are not adopted to measure the differences between the proposed method and SOTA methods, such as Disentanglement, Completeness, Informativeness， et al.  [1]
2. Some related multi-dimensional disentangling methods (which are not probabilistic)  with AE as backbone are not included [2,3].
3. The proposed method is only validated on the toy / synthetic dataset. The author is suggested to add experiments on natural image datasets.
4. Some typos, such as ' of each of', 'commutative Lie groups'
5. I am concerned that the proposed method will fail on natural images. The reasons are concluded as follows: the PCA achieves poor results on complicated nature images; The nature images contain more noise.

References:
[1]A Framework for the Quantitative Evaluation of Disentangled Representations[J]. International Conference on Learning Representations, 2018.
[2]Dual Swap Disentangling, NIPS 2018.
[3]Image-to-image translation for cross-domain disentanglement, NIPS 2018.



**Summary Of The Paper:**

In this paper, the author proposed a new perspective for learning disentangled representation. The Euler encoding is introduced to force the latent space to achieve a linear disentangled representation. What's more, the author adopts the PCA, normalization layer, and Gaussian interpolation to address the issues caused by the Euler encoding. The author also designed a new metric for comparing the disentanglement of the representation. The proposed method is validated on several toy datasets.


**Summary Of The Review:**

Overall, the ideal is novel. The author provided a new insight for representation learning with Euler encoding.
However, the experiments on the nature image dataset are not given. Some classic metrics are not adopted in the comparison experiments.

---

> ### Author Response · Authors · 2022-11-13
> **Response to Reviewer JNBF**
>
> 1. Metrics
>
> We agree that there is a wide range of metrics to measure disentanglement, completeness and informativeness. In fact, we used the metric proposed in [1], which we call dci-rf (Disentanglement, Completeness and Informativeness using random forest regressor). We have now made this clear in our paper.
>
> 2. Other model comparison
>
> Thank you for the references. We agree that there are still many disentangling methods, but, it is difficult to compare all of them. Therefore, we focus on models that do not require any labels or pairs of inputs and are entirely autoencoder-based. Since these papers require a small number of labels or pairs of inputs, comparing the proposed model against two methods go beyond our target. We have made this clearer in our modified version of the paper.
>
> 3. Experiments on natural image datasets.
>
> Thank you for the suggestion. We have now added a new result from blood cell dataset [1] in the Appendix due to the space limit (see Figure 24). Although this dataset does not have desirable group structures, there are two key factors that can help assessing the quality of disentanglement: (1) size and (2) shape of blood cells, both are essential for classifying the blood cells as a downstream task. We visually show that the proposed models learn size and segmentation process of cells independently. In addition, we also have added the Sprites dataset [2] in the Appendix (see Figure 23), which is more complex and other papers submitted to this ICLR 2023 conference used.
>
> [1] A dataset of microscopic peripheral blood cell images for development of automatic recognition systems," Data in Brief, vol. 30, pp. 105474, 2020.
>
> [2] Deep visual analogy-making. Advances in neural information processing systems, 28, 2015.
>
> 4. Some typos
>
> Thank you for pointing out. These all have been fixed now.
>
> 5. About PCA and  nature images
>
> We used the PCA technique as a part of obtaining the hyper-parameter, and PCA is not the only algorithm we can use. In fact, the model can accommodate any valid replacement to PCA, such as VAE variants, to extract the variance information from compressed information. Additionally, we have now added a new result from blood cell dataset, which is not synthetic data, and the result shows that we still can learn key features.

---

### Author Response · Authors · 2022-11-13
**General response**

Thank you for all the reviews, providing very useful comments and suggestions on the manuscript. We have revised the manuscript, addressing each concern from the reviewers very carefully. There are a few comments that came up from different reviewers. We would like to make some general comment, which would benefit all reviews.

1. Evaluation based on real-world images:
Although real-world datasets, such as 3D Chair (Burgess & Kim, 2018) and CelebA (Liu et al., 2015), can be found in the literature, we have not used them in our work as they lack a clear underlying group structure. However, we have added a new real-world dataset, namely, Blood Cell dataset [1] in the Appendix due to the space limit (see Figure 24), to our evaluation. Although this dataset lacks desirable levels of group structures, there are two key factors that can help assessing the quality of disentanglement: (1) size and (2) shape of blood cells – both are essential for a number of downstream tasks, such as classification of blood cells. We visually show that the proposed models learn size and segmentation process of cells independently. In addition, we also have added the Sprites dataset [2] in the Appendix (see Figure 23), which is more complex and other papers submitted to this ICLR 2023 conference used.

2. Comparing against other models.
We agree that there are many disentangling methods in the literature along with a large number of possible metrics. However, understandably, it is a monumental task to compare all of them let alone presenting the results. For practical reasons, we limited our evaluation to against ten other models, which are either autoencoder-based probabilistic models or symmetry-based disentangled models, that do not require any labels or pairs of inputs. This approach makes the models impartial, and directly relevant to the proposed approach. We have now made this clear in our modified version of the manuscript.

[1] A dataset of microscopic peripheral blood cell images for development of automatic recognition systems," Data in Brief, vol. 30, pp. 105474, 2020.

[2] Deep visual analogy-making. Advances in neural information processing systems, 28, 2015.

---

> ### Author Response · Authors · 2022-11-17
> **Supplementary Material update**
>
> We have uploaded a diff file as supplementary material.

---

### Decision · Program_Chairs · 2023-01-20

**Decision:**

Reject

**Justification For Why Not Higher Score:**

Regarding the accepted papers in my batch this falls under borderline reject, although the approach seems interesting from the abstract it also does not appear as impactful to neglect the 3 and 5 scores. Also, most disentanglement methods do not require paired samples, e.g. betaVAE etc.


**Justification For Why Not Lower Score:**

N/A

**Metareview: Summary, Strengths And Weaknesses:**

The work presents a deterministic model for linear disentanglement through Euler encoding that does not require pair of images or labels as some of the related works. The concern on toy and synthetic datasets has been partially addressed and does not constitute a major issue for this type of works, however although the method is considered novel the weaknesses at the moment outweighs the strengths.
Excellent work has been done already to address the reviewers' concerns and I am confident about the potential of the approach, I therefore strongly encourage the authors to further improve their work and resubmit.